



# WFDE5: bias adjusted ERA5 reanalysis data for impact studies

Marco Cucchi[1], Graham P. Weedon[2], Alessandro Amici[1], Nicolas Bellouin[3], Stefan Lange[4],
Hannes Müller Schmied[5,6], Hans Hersbach[7], and Carlo Buontempo[7]

[1]B-Open Solutions srl, Rome, Italy
[2]Met Office, Maclean Building, Benson Lane, Crowmarsh Gifford, Wallingford, Oxfordshire, OX10 8BB, United Kingdom
[3]Department of Meteorology, University of Reading, Reading, RG6 6BB, United Kingdom
[4]Potsdam Institute for Climate Impact Research (PIK), Member of the Leibniz Association, P.O. Box 60 12 03, 14412
Potsdam, Germany
[5]Institute of Physical Geography, Goethe University Frankfurt, Frankfurt am Main, Germany
[6]Senckenberg Leibniz Biodiversity and Climate Research Centre (SBiK-F), Frankfurt am Main, Germany
[7]European Centre for Medium-Range Weather Forecasts, Reading, United Kingdom

**Correspondence:** Carlo Buontempo (carlo.buontempo@ecmwf.int)

**Abstract.** The WFDE5 dataset (C3S, 2020) has been generated using the WATCH Forcing Data (WFD) methodology applied to surface meteorological variables from the ERA5 reanalysis. The WFDEI dataset had previously been generated by applying the WFD methodology to ERA-Interim. The WFDE5 is provided at 0.5° spatial resolution, but has higher temporal resolution (hourly) compared to WFDEI (3-hourly). It also has higher spatial variability since it was generated by aggregation of the

higher-resolution ERA5 rather than by interpolation of the lower resolution ERA-Interim data. Evaluation against meteorological observations at 13 globally distributed FLUXNET2015 sites shows that, on average, WFDE5 has lower mean absolute error and higher correlation than WFDEI for all variables. Bias-adjusted monthly precipitation totals of WFDE5 results in more plausible global hydrological water balance components as analyzed in an uncalibrated hydrological model (WaterGAP) than use of raw ERA5 data for model forcing.

The dataset, which can be downloaded from https://doi.org/10.24381/cds.20d54e34 (C3S, 2020), is distributed by the Copernicus Climate Change Service (C3S) through its Climate Data Store (CDS, Copernicus Climate Change Service, 2020) and currently spans from the start of January 1979 to the end of 2018. The dataset has been produced using a number of CDS Toolbox applications, whose source code is available with the data - allowing users to re-generate part of the dataset or apply the same approach on other data. Future updates are expected spanning from 1950 to the most recent year.

A sample of the complete dataset, which covers the whole 2016 year, is accessible without registration to the CDS at https://doi.org/10.21957/935p-cj60.



# 1 Introduction

The development, calibration and evaluation of impact models requires good quality historical meteorological datasets. These are needed to both drive the impact models themselves and characterise their performances over the historical period. The availability of reliable historical runs is also critical for the preparation of impact studies using climate projections. Reanalyses have long been used for those purposes as they provide a physically consistent global reconstruction of past weather without any gap in space or in time. The ERA-Interim global reanalysis for the atmosphere, land surface and ocean waves (Dee et al., 2011) of the European Centre for Medium Range Weather Forecast (ECMWF) has been used widely as a reference by the climate community. Although reanalyses represent -by construction- the most plausible state of the atmosphere and the ocean given the observations and the forecasts from the model at a previous time-step, the coarse resolutions of models, the assumptions made in sub-grid parameterisations, and more generally the overall inadequacies of the modelling framework, are known to induce biases with respect to ground-based observations and radiosondes. Considering that the primary goal of impact studies is to assess the climate change impacts in the real world (as opposite to the model world) it is essential that such biases are first characterised and then, as much as practically possible, corrected for.

Recently the ERA5 reanalysis has superseded the ERA-Interim reanalysis (Hersbach et al., 2020 under review). It is produced at ECMWF as part of the EU-funded Copernicus Climate Change Service (C3S). At the time of writing data was available from the C3S Climate Data Store (CDS) for the period from 1979 onwards. Timely updates are provided with a 5-day latency, while a more thorough quality check is provided 2-3 months later. In 2020 the dataset will be extended back to 1950, and will then also encompass the period covered by ERA-40 (1957–2002). ERA5 is based on 4D-Var data assimilation using Cycle 41r2 of the Integrated Forecasting System (IFS), which was operational at ECMWF in 2016. As such, compared to ERA-Interim (which was based on an IFS cycle that dates from 2006) ERA5 benefits from a decade of developments in model physics, core dynamics and data assimilation. In addition to a significantly enhanced horizontal resolution (31 km grid spacing compared to 80 km for ERA-Interim), ERA5 has a number of innovative features. These include hourly output throughout and an uncertainty estimate. The uncertainty information is obtained from a 10-member ensemble of data assimilations with 3-hourly output at half the horizontal resolution (63 km grid spacing). Compared to ERA-Interim, ERA5 also provides an enhanced number of output parameters. The move from ERA-Interim to ERA5 represents a step change in overall quality and level of detail. An overview of the main characteristics and general performance of ERA5 and a comparison with ERA-Interim is provided in Hersbach et al. (2020 under review), while more in-depth studies of particular aspects have been reported in a growing number of publications in the scientific literature.

ERA5 is based on a vast amount of synoptic observations. The number has increased from approximately 0.75 million per day on average in 1979 to around 24 million per day by the end of 2018. Satellite radiances are the dominant and growing type of data throughout the period. The volume of conventional data has also increased steadily. In addition to observations, ERA5 relies on gridded information about radiative forcing and boundary conditions. For radiation, ERA5 includes forcings for total solar irradiance, ozone, greenhouse gases and some aerosols developed for the World Climate Research Programme (WCRP) initiative CMIP5, including stratospheric sulphate aerosols. This represents a major improvement on ERA-Interim, which, for





example, does not account for stratospheric sulphate aerosols due to major volcanic eruptions. Details are provided in Hersbach et al. (2015). The evolution of sea-surface temperature (SST) and sea ice cover is based on a combination of products: the UK Met Office Hadley Centre HadISST2 product for SST, the EUMETSAT OSI-SAF reprocessed product for sea ice, and the UK Met Office OSTIA product for SST and sea ice that is also used in ECMWF's operational forecasting system. Details can be

found in Hirahara et al. (2016).

The EU WATCH programme produced a common framework for land surface models (LSMs) and global hydrological models (GHMs) to assess the global terrestrial hydrological cycle in the 20th & 21th centuries. This required a common meteorological forcing dataset for the 20th century which became the WATCH Forcing Data (WFD). The WFD, based on the ERA40 reanalysis (Uppala et al., 2005), allowed intercomparisons of hydrological models and bias correction of 21st century

GCM outputs (Haddeland et al., 2011; Hagemann et al., 2011). The modelling in WATCH required sub-daily, and daily average data, at half-degree spatial resolution necessitating interpolation onto the regular latitude-longitude grid, land-sea mask and elevations used by the Climate Research Unit (CRU). The WFD methodology (Weedon et al., 2010, 2011) involved common processing of all terrestrial half-degree grid boxes outside Antarctica at three hourly steps, with elevation correction of air temperature and consequent adjustment of surface pressure, specific humidity and downwards longwave radiation. Bias correction

utilized the CRU gridded observations (New et al., 1999, 2000) of monthly average air temperature, diurnal temperature range, cloud cover (for adjusting average downwards shortwave fluxes), precipitation totals and number of "wet" (i.e. precipitation) days. Additionally, downwards shortwave radiation was corrected for changes in multi-year tropospheric and stratospheric aerosol loading. Unlike most other reanalyses ERA provides rainfall and snowfall rates separately and this permitted adjustment of these rates to allow for the precipitation gauge catch corrections inherent in the observed CRU precipitation totals.

Though critical for hydrological modelling, the precipitation variables are the least well constrained by surface observations, so data were provided in two versions dependent on the source of the gridded monthly observed precipitation totals – one based on CRU and the other on the "full data product" of the Global Precipitation Climatology Centre (GPCC).

Later the WFD methodology was applied to the ERA-Interim reanalysis (Dee et al., 2011) to produce the WFDEI dataset (Weedon et al., 2014). As before the reanalysis data were 3-hourly and interpolated onto the CRU land-sea mask. Unlike

the WFD, the WFDEI includes Antarctica and an extra processing step was introduced for the precipitation variables after correction of monthly totals and numbers of wet days, and before correction of precipitation gauge biases. This involved overriding the reanalysis ratio of rainfall to snowfall in each time step in cases where the differences between the CRU grid box elevation differed substantially from ERA-Interim elevation (Weedon et al., 2014). Intermittent updates of the WFDEI beyond 2009 used the latest versions of CRU and GPCC – i.e. WFDEI files for additional years were added rather than entire new

versions of the files created.

Here a new dataset is described based on the ERA5 reanalysis: the WFDE5 (i.e. "WATCH Forcing Data methodology applied to ERA5 reanalysis data", C3S, 2020). In this case the data are available at hourly instead 3-hourly steps and the higher resolution of ERA5 required aggregation to half-degree spatial resolution instead of interpolation. In addition, as described later, the aerosol correction step for downwards shortwave radiation has been revised. Bias correction involved the CRU TS4.03

data for 1979-2018 inclusive and the alternative precipitation totals based on the full data monthly GPCCv2018 product for



**Table 1.** Sources of data used to derive the WFDE5 dataset

| Dataset | Summary | Location |
|---------|---------|----------|
| ERA5 | ECMWF reanalysis product | https://cds.climate.copernicus.eu/cdsapp#!/home |
| CRU TS4.03 | Climate Research Unit gridded station observations (multiple variables) | http://data.ceda.ac.uk/badc/cru/data/cru_ts/cru_ts_4.03 |
| GPCCv2018 | Global Precipitation Climatology Centre gridded station precipitation observations | https://opendata.dwd.de/climate_environment/GPCC/html/fulldata-monthly_v2018_doi_download.html |

rainfall rates and snowfall rates for 1979-2016 inclusive (Harris et al., 2014; Schneider et al., 2017). For an outline of the methodology applied and a reference to the observation datasets used see Tables 1 and 2.

WFDE5 is a meteorological forcing dataset for land surface and hydrological models. It consists of eleven variables (see Table 2) with an hourly temporal resolution on a regular longitude-latitude half-degree grid, with global spatial coverage
and values defined only for land and lake points. The dataset was derived by applying sequential elevation and monthly bias correction methods described in Weedon et al. (2010, 2011) to half-degree aggregated ERA5 reanalysis products (Copernicus Climate Change Service, 2017). The monthly observational datasets used for bias correction are CRU TS4.03 from CRU (Harris et al., 2014) for 1979 to 2018 for all variables and the GPCCv2018 full data product (Schneider et al., 2018) for rainfall and snowfall rates for 1979 to 2016.

As a meteorological forcing dataset, WFDE5 facilitates climate impact simulations such as those carried out in the Inter-Sectoral Impact Model Intercomparison Project (ISIMIP; Warszawski et al., 2014; Frieler et al., 2017). It can be used to directly drive historical impact simulations, which are needed for impact model validation. It can also be used as an observational reference dataset for the bias adjustment of future climate projections; these bias-adjusted climate projections can then be used to drive future climate impact projections. Both predecessors of WFDE5 have been employed for these two purposes
in previous ISIMIP phases. In particular, the bias adjustment of future climate projections was done using the WFD in the ISIMIP Fast Track (Hempel et al., 2013) and the EartH2Observe, WFDEI and ERA-Interim data Merged and Bias-corrected for ISIMIP (EWEMBI; Lange, 2018, 2019a) in ISIMIP2b (Frieler et al., 2017). WFDE5 will be similarly employed in the upcoming ISIMIP phase 3.

## 2 Dataset Processing

All computations were carried out within the CDS Toolbox, a python coding environment to retrieve, process, plot and download data from the C3S Climate Data Store (CDS, Copernicus Climate Change Service, 2020). The CDS Toolbox scripts used to generate the dataset are publicly available at https://doi.org/10.24381/cds.20d54e34 under a free and open licence, and can be used to reproduced the dataset.



**Table 2.** WFDE5 elevation and bias correction methodology outline (Weedon et al., 2010, 2011)

| Variable name | Description | Units | Time step adjustments | Data used for monthly bias correction |
|---|---|---|---|---|
| Wind | 10 m wind speed | m s$^{-1}$ | Nil | Nil |
| Tair | 2 m air temperature | K | Via environmental lapse rate | CRU TS4.03 temperature and diurnal temperature range |
| PSurf | Pressure at the surface | Pa | Via changes in Tair | Nil |
| Qair | 2 m specific humidity | kg kg$^{-1}$ | Via changes in Tair and PSurf | Nil |
| LWdown | Downward longwave radiation flux | W m$^{-2}$ | Via fixed relative humidity and changes in Tair, PSurf, and Qair | Nil |
| SWdown | Downward shortwave radiation flux | W m$^{-2}$ | Nil | CRU TS4.03 cloud cover and effects of interannual changes in atmospheric aerosol loading |
| Rainf $^{(CRU)}$ | Rainfall rate | kg m$^{-2}$ s$^{-1}$ | Adjustment of snow/rainfall ratios | CRU TS4.03 number of wet days, CRU TS4.03 precipitation totals, ERA5 ratio of rainfall/precipitation, rainfall gauge correction |
| Snowf $^{(CRU)}$ | Snowfall rate | kg m$^{-2}$ s$^{-1}$ | Adjustment of snow/rainfall ratios | CRU TS4.03 number of wet days, CRU TS4.03 precipitation totals, ERA5 ratio of rainfall/precipitation, snowfall gauge correction |
| Rainf $^{(CRU+GPCC)}$ | Rainfall rate | kg m$^{-2}$ s$^{-1}$ | Adjustment of snow/rainfall ratios | CRU TS4.03 number of wet days, GPCCv2018 precipitation totals, ERA5 ratio of rainfall/precipitation, rainfall gauge correction |
| Snowf $^{(CRU+GPCC)}$ | Snowfall rate | kg m$^{-2}$ s$^{-1}$ | Adjustment of snow/rainfall ratios | CRU TS4.03 number of wet days, GPCCv2018 precipitation totals, ERA5 ratio of rainfall/precipitation, snowfall gauge correction |
| ASurf | Grid-points altitude | m | Nil | Nil |

NOTE: Variable names and units are based on the ALMA (Assistance for Land-surface Modeling Activities) conventions (http://www.lmd.jussieu.fr/ polcher/ALMA/); Wind, Tair, PSurf and Qair variables have instantaneous values, while LWdown, SWdown, Rainf and Snowf have average over the next hour at each date-time.



**Table 3.** Mapping between CDS validity date-time and forecast base date-time and step. "date-1" refers to the previous day.

| Validity | | Forecast | | | Validity | | Forecast | | | Validity | | Forecast | | |
| date | time | date | time | step | date | time | date | time | step | date | time | date | time | step |
|---|---|---|---|---|---|---|---|---|---|---|---|---|---|---|
| date | 00 | date-1 | 18 | 06 | date | 08 | date | 06 | 02 | date | 16 | date | 06 | 10 |
| date | 01 | date-1 | 18 | 07 | date | 09 | date | 06 | 03 | date | 17 | date | 06 | 11 |
| date | 02 | date-1 | 18 | 08 | date | 10 | date | 06 | 04 | date | 18 | date | 06 | 12 |
| date | 03 | date-1 | 18 | 09 | date | 11 | date | 06 | 05 | date | 19 | date | 18 | 01 |
| date | 04 | date-1 | 18 | 10 | date | 12 | date | 06 | 06 | date | 20 | date | 18 | 02 |
| date | 05 | date-1 | 18 | 11 | date | 13 | date | 06 | 07 | date | 21 | date | 18 | 03 |
| date | 06 | date-1 | 18 | 12 | date | 14 | date | 06 | 08 | date | 22 | date | 18 | 04 |
| date | 07 | date | 06 | 01 | date | 15 | date | 06 | 09 | date | 23 | date | 18 | 05 |

## 2.1 Extraction and aggregation of reanalysis data

ERA5 reanalysis data are available in the CDS on regular latitude-longitude grids at $0.25° \times 0.25°$, as a result of finite element-based linear interpolation from original reduced Gaussian grid at $\sim 0.28°$, and atmospheric parameters are distributed on 37 pressure levels. They are distributed at hourly resolution as analyses, for instantaneous variables, or forecasts, for accumulated variables. The date and time of the data is specified using the validity date-time, so step does not need to be specified. For forecasts, steps between 1 and 12 hours have been used to provide data for all the validity times in 24 hours (Table 3).

Accumulation variables are aggregated over the hour ending at the forecast step, but they are automatically converted to mean rates when retrieved from within the CDS Toolbox.

Before applying elevation and bias correction, two preprocessing steps were performed on ERA5 reanalysis data. First, in order to enable comparison and bias correction using the CRU dataset, ERA5 reanalysis were regridded to regular half-degree longitude-latitude grid, via first-order conservative remapping (Jones, 1999). Then, a backward one-hour time shift was applied

to rate variables, so that values stored at each date-time represents time averages over the following hour. The latter step was taken in order to adhere to the scheme used for the WATCH Forcing Dataset (Weedon et al., 2011).

It is worth noticing that grid-points classified as belonging to land in CRU TS4.03 and GPCCv2018 datasets are not necessarily classified as land-points in ERA5 reanalysis dataset. This is especially true for coastal grid-points, for which not considering this issue often led to anomalous values in the first iteration of the WFDE5 dataset. For this reason, besides

applying CRU TS4.03/GPCCv2018 to ERA5 reanalysis after half-degree regridding, an additional mask derived by ERA5 quarter-degree land-sea and lake cover mask is applied just after retrieval. In this way, the final WFDE5 dataset contains values only for all grid-points which are classified as land or lake by both ERA5 and CRU.

## 2.2 Elevation and bias correction

Once aggregation had been performed, the sequential elevation and monthly bias correction methods of Weedon et al. (2010,

2011) were applied to the regridded data (see Table 2). The same procedures used for the creation of the WFDEI (Weedon et al., 2014) were applied, with the only exception of near-surface specific humidity (Qair). For this variable, given the absence





of both ERA5 near-surface specific and relative humidity from the CDS, a slightly different approach was taken: first, ERA5 vapor pressure and saturation vapor pressure at the surface, $e$ and $e_{sat}$ respectively, were computed following Buck (1981); then, they were used to compute ERA5 relative humidity at surface as $RH = 100.0 \cdot e/e_{sat}$; finally, at this point, the algorithm described in Weedon et al. (2010) could be resumed.

Likewise for WFD (Weedon et al., 2011) and WFDEI (Weedon et al., 2014) datasets, downward shortwave radiation was adjusted at the monthly time scale using CRU cloud cover and the local linear correlation between monthly average (aggregated) ERA5 cloud cover and downward shortwave radiation (Sheffield et al., 2006; Weedon et al., 2010).

ERA5 includes a simplified representation of the time evolution of sulfate aerosols, which interact with radiation only in that model, but otherwise does not account for the impact on surface radiative fluxes of changes in aerosol interactions with radiation (also called direct effects of aerosols) and clouds (also called first indirect effects of aerosols). To represent those impacts, aerosol corrections are calculated as monthly distributions of the anomaly in downward surface shortwave radiative flux due to aerosol-radiation and aerosol-cloud interactions over the period 1979-2018. Radiative transfer calculations, which use the tools described in section 2.f.ii of Weedon et al. (2010), are based on monthly-averaged distributions of tropospheric and stratospheric aerosol optical depth, and cloud fraction. The time series of tropospheric optical depth for sulfate, fossil-fuel black and organic carbon, biomass burning, mineral dust, seasalt, and secondary biogenic aerosols is taken from the historical and RCP8.5 simulations by the HadGEM2-ES climate model (Bellouin et al., 2011). To correct for biases in HadGEM2-ES aerosol optical depths, these optical depths are scaled over the whole period and for each aerosol species to match the global and monthly averages obtained by the CAMS Reanalysis of atmospheric composition (2003-2017; Inness et al. (2019)), which assimilates satellite retrievals of aerosol optical depth. This bias correction was not applied in WFD and WFDEI but is now possible thanks to the availability of the CAMS Reanalysis. The time series of stratospheric aerosol optical depth is taken from the climatology by Sato et al. (1993), which has been updated to 2012 at data.giss.nasa.gov/modelforce/strataer/. Years 2013-2017 are assumed to match background years so they replicate year 2010. That assumption is supported by the Global Space-based Stratospheric Aerosol Climatology time series (1979-2016; Thomason et al., 2018). The time series of cloud fraction is taken from CRU TS 4.03, for consistency with other aspects of the WFDE5 dataset. Surface radiative fluxes account for aerosol-radiation interactions from both tropospheric and stratospheric aerosols, and for aerosol-cloud interactions from tropospheric aerosols, except mineral dust. The radiative effects of aerosol-cloud interactions are assumed to scale with the radiative effects of aerosol-radiation interactions, using regional scaling factors derived from HadGEM2-ES. To avoid double-counting the radiative effects of aerosol-radiation interactions by sulfate aerosols, which are to some extent already represented in ERA5, the radiative transfer calculations are repeated, this time only including sulfate aerosol-radiation interactions, and the corresponding anomalies subtracted from the set of fluxes obtained previously. Atmospheric constituents other than aerosols and clouds are set to a constant standard mid-latitude summer atmosphere, because their variations only have second-order effects on aerosol corrections.

Finally, similarly to the WFD and WFDEI datasets, two different WFDE5 rainfall and snowfall rates datasets, including gauge catch corrections, were generated by using either CRU TS4.03 or GPCCv2018 precipitation totals. The GPCCv2018 database includes around 3–4 times as many precipitation stations as CRU (incorporating most of the latter as a subset (Becker



**Table 4.** Summary of WFDE5 dataset attributes on the C3S Climate Data Store

| Dataset attribute | Details |
|---|---|
| Horizontal coverage | Global |
| Horizontal resolution | 0.5° x 0.5° |
| Vertical coverage | Surface |
| Temporal coverage | - 1979-01-01 00:00:00 to 2018-12-31 23:00:00 for variables Wind, Tair, PSurf and Qair <br> - 1979-01-01 07:00:00 to 2018-12-31 23:00:00 for variables LWdown, SWdown, Rainf[CRU], Snowf[CRU] <br> - 1979-01-01 07:00:00 to 2016-12-31 23:00:00 for variables Rainf[CRU+GPCC], Snowf[CRU+GPCC] |
| Temporal resolution | Hourly |
| File format | NetCDF |
| Data type | Grid |
| Version | 1.0 |
| File naming convention | `<var>_WFDE5_<reference_dataset>_<YYYYMM>_v1.0.nc`, <br> where <br> - `<var>`: variable name, as in Table 2 <br> - `<reference_dataset>`: one between CRU (all variables) and CRU+GPCC (Rainf and Snowf only) <br> - `<YYYYMM>`: year and month |

et al., 2013; Schneider et al., 2014)), but extends only till 2016. As already pointed out in Weedon et al. (2014), during generation of the WFDE5 precipitation rates an error in the precipitation phase can arise locally where there are large elevation differences between ERA5 and CRU grids. For this reason, a further processing step was added to the WFD methodology to

175 correct the most extreme cases of inappropriate precipitation phase: for each grid box and each calendar month over 1979–2018, records of the minimum Tair during rainfall and the maximum Tair during snowfall ("phase temperature extremes") were stored; then, for each grid box and hourly time step, the precipitation phase was switched if the combination of the phase with the elevation and bias-corrected Tair laid beyond a phase temperature extreme.

Elevation and bias correction was applied for all land points outside Antarctica. For grid points belonging to this region,

given the absence of observational data, only elevation correction was applied.

## 3   Availability and access

WFDE5 dataset is distributed through the C3S Climate Data Store as monthly files in NetCDF format, and can be downloaded at https://doi.org/10.24381/cds.20d54e34 (C3S, 2020). It uses a full half-degree grid (720 × 360 grid boxes) with the sea/large lakes flagged as missing data, comprising a total of 92889 land points (Antarctica included). General dataset attributes are



described in Table 4. A sample of the complete dataset, which covers the whole 2016 year, is accessible without registration to the CDS at https://doi.org/10.21957/935p-cj60.

All the CDS Toolbox workflows used to generate WFDE5 are publicly available https://doi.org/10.24381/cds.20d54e34, and can be used to re-generate samples of the dataset. Furthermore, as ERA5 progresses, using these applications it will be possible to expand WFDE5 dataset back to the start of 1950 and forward beyond 2018.

## 4  Evaluation

### 4.1  Previous analyses

Beck et al. (2019a) assessed multiple precipitation datasets at daily time steps against radar and precipitation gauge observations across the co-terminus USA. Their analysis included ERA5, ERA-Interim and WFDEI precipitation adjusted to GPCC totals. They demonstrated that against observations ERA5 precipitation provides a significant improvement over both ERA-
Interim and WFDEI precipitation. Albergel et al. (2018) used the ISBA LSM to assess the use of ERA5 versus ERA-Interim forcing. They assessed performance against a wide variety of observed hydrological and vegetation-related variables. Significant improvements were demonstrated in simulation of the hydrological cycle using ERA5 which they mostly attributed to better precipitation. There were small changes related to vegetation modelling. For a region with a low density of gauges in Iran, Fallah et al. (2020) showed that ERA5 precipitation is closer to local observations than ERA-Interim but that GPCCv8
(used here in bias correction or ERA5) is substantially better.

### 4.2  Comparison with FLUXNET2015 and WFDEI

The FLUXNET2015 (FN2015) meteorological data (Chu, 2015; Pastorello et al., 2017) are not included in the data assimilation of the ERA5 reanalysis. Therefore, these data provide an opportunity to assess the degree to which the ERA5 and WFDE5 meteorological variables agree with surface observations. Despite there being over two hundred FN2015 sites globally they
are highly clustered within Europe and North America. In order to provide a fairly uniform global assessment, 13 sites with at least three years of data, have been selected from 12 countries spanning a wide range of longitudes and latitudes (Fig. 1, Table 5). The primary purpose of the FN2015 meteorological dataset is to provide data for forcing LSMs to allow comparison with the FN2015 surface exchange fluxes of energy and carbon. As such the FN2015 meteorological variables have been gap filled using ERA-Interim data to allow modelling without missing data. To avoid biasing the comparisons made here, only
meteorological values that are measurements have been used (i.e. at times and locations where the FN2015 tier 1 quality flag is 0). Unfortunately, this means that some FN2015 sites do not provide observations for some variables at any time steps ("Missing variables" in Table 5).

Two pairs of comparisons have been made: firstly ERA5 (aggregated to half degree) versus FN2015 and WFDE5 versus FN2015 at an hourly time step. This required converting the half-hourly FN2015 data to hourly steps and aligning the time
stamps since ERA5 is on UTC instead of local time. ERA5 does not provide specific humidity so Qair was calculated using the

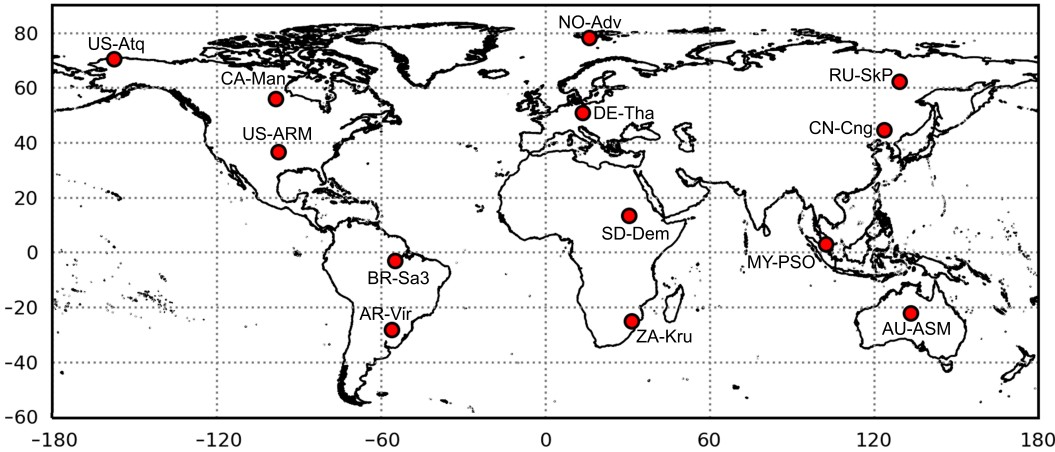

**Figure 1.** Location of FLUXNET2015 sites used to evaluate ERA5, WFDE5 and WFDEI.

**Table 5.** Selected FLUXNET2015 sites.

| Site code | Country | Site name | Longitude | Latitude | Start year | End year | Missing variables | DOI |
|---|---|---|---|---|---|---|---|---|
| US-Atq | USA | Atqauk | 157.41ºW | 70.47ºN | 2006 | 2008 | LWdown | 10.18140/FLX/1440067 |
| CA-Man | Canada | Manitoba | 98.48ºW | 55.88ºN | 2006 | 2008 | PSurf, LWdown | 10.18140/FLX/1440035 |
| US-ARM | USA | ARM Southern Great Plain site - Lamont | 97.49ºW | 36.61ºN | 2010 | 2012 | | 10.18140/FLX/1440066 |
| BR-Sa3 | Brazil | Santarem km67 primary forest | 54.97ºW | 3.02ºN | 2001 | 2003 | | 10.18140/FLX/1440033 |
| AR-Vir | Argentina | Virasoro | 56.19ºW | 28.24ºS | 2010 | 2012 | LWdown, Precip | 10.18140/FLX/1440192 |
| NO-Adv* | Norway | Adventdalen | 15.92ºE | 78.19ºN | 2012 | 2014 | | 10.18140/FLX/1440241 |
| DE-Tha | Germany | Tharandt | 13.57ºE | 50.96ºN | 2012 | 2014 | | 10.18140/FLX1440152 |
| SD-Dem | Sudan | Demokeya | 30.48ºE | 13.28ºN | 2007 | 2009 | LWdown | 10.18140/FLX/1440186 |
| ZA-Kru | South Africa | Skukuza | 31.50ºE | 25.02ºS | 2008 | 2010 | PSurf, LWdown | 10.18140/FLX/1440188 |
| RU-SkP | Russia | Yakutsk Spasskaya Pad larch | 129.17ºE | 62.26ºN | 2010 | 2014 | Precip | 10.18140/FLX/1440243 |
| CN-Cng | China | Chanling | 123.51ºE | 44.59ºN | 2008 | 2010 | | 10.18140/FLX/1440209 |
| MY-PSO | Malaysia | Pasoh Forest Reserve | 102.31ºE | 2.97ºN | 2007 | 2009 | PSurf | 10.18140/FLX/1440240 |
| AU-ASM | Australia | Alice Springs | 133.25ºE | 22.28ºS | 2011 | 2013 | | 10.18140/FLX/1440194 |

*NO-Adv is now designated as SL-Adv (i.e. within Svalbard). Precip = precipitation. Note that "Missing variables" refers to tier 1 items provided by FLUXNET2015 as entirely gap-filled, not measured, values.

2 m air temperature, surface pressure and relative humidity using equations 4 and 6 of Buck (1981). The second comparisons were for WFDEI versus FN2015 and for WFDE5 versus FN2015 at 3-hourly time steps, again with alignment of time stamps. At each site mean bias error (MBE), mean absolute error (MAE) and correlation were calculated. MAE was used instead of root mean square error since the former provides a less ambiguous basis for assessment (Willmott and Matsuura, 2005). Since

the data are time series there is considerable serial correlation leading to spuriously high values. Consequently, the correlations of the previously pre-whitened time series – i.e. adjusted for lag-1 autocorrelation - are also reported as "adjusted correlation" (Ebisuzaki, 1997). Data for individual sites are reported for ERA5 and WFDE5 v FN2015 (hourly) in Tables A1 to A8 and for WFDEI and WFDE5 v FN2015 (3 hourly) in Tables A9 to A16. Average metrics for the pairs of comparisons are shown in Fig. 2 and Tables A18 and A17.



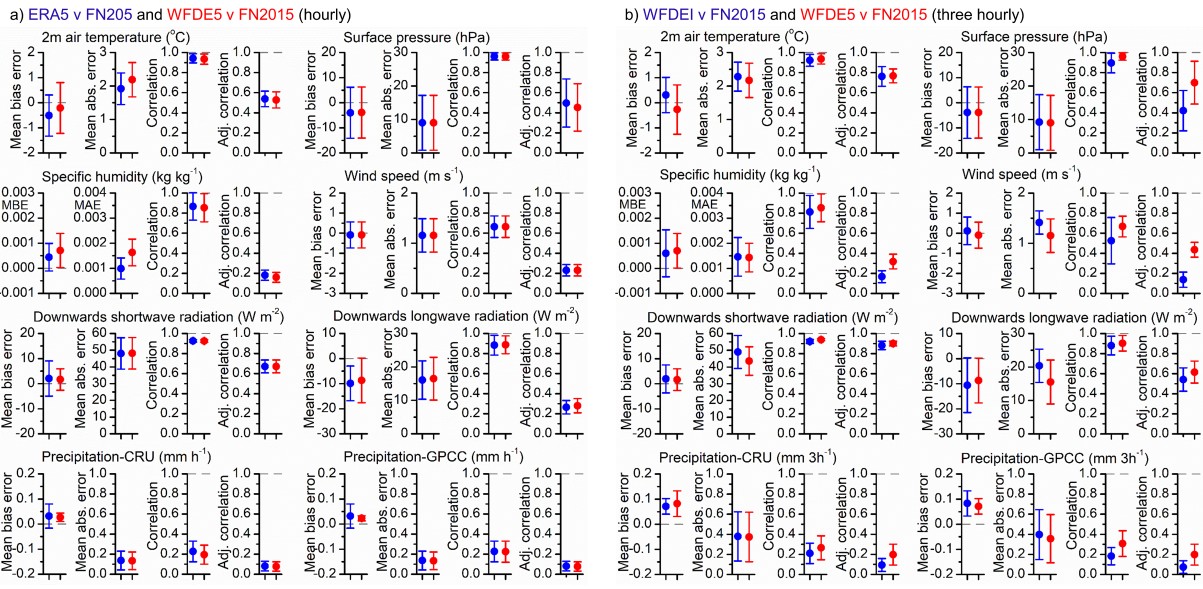

**Figure 2.** Average metrics (mean +/- 95% confidence interval of the mean) for ERA5 v FN2015 (blue) and WFDE5 v FN2015 (red) at hourly time steps (see Table A17).

At hourly steps on average there are no significant differences in MBE, MAE, correlation or adjusted correlation between ERA5 v FN2015 and WFDE5 v FN2015, for all variables apart from two (Fig. 2a). For air temperature the MBE is slightly better (closer to zero) for WFDE5 whereas the MAE is slightly worse (larger) for WFDE5. On the other hand, for specific humidity both the MBE and MAE are slightly worse for WFDE5. These results indicate that the bias and elevation corrections incorporated into the WFDE5 have had little overall effect on the performance against surface observations compared to ERA5.

At three hourly steps, for all variables apart from precipitation, the average MBE overlaps zero for WFDEI and WFDE5 (Fig. 2b). For wind speed, downwards longwave and downwards shortwave the MAE is slightly better (smaller) for WFDE5 than WFDEI. For all variables, aside from precipitation, the MAE, correlation and adjusted correlation are slightly better for WFDE5 than WFDEI. For precipitation the MBE is slightly better and the correlation slightly higher for WFDE5 versus WFDEI when corrected using the GPCC-, rather than CRU-precipitation totals. These results indicate that on average, at the

FN2015 sites selected, WFDE5 performs better than WFDEI against the observations. Note that the average results in Fig. 2b and Table A17 hide the fact that for all metrics WFDEI data provide better results (MBE closer to 0.0, MAE lower, correlation higher) for some individual sites than WFDE5 (Tables A9 to A16). On the other hand, for Wind (speed) and Precipitation (CRU and GPCC corrected) the correlation and adjusted correlation is better for WFDE5 than WFDEI at every site.

    Both WFDEI and WFDE5 in 2017 and 2018 are corrected using CRU TS4.03 so at monthly and longer scales there will

be only small differences. However, at sub-monthly time scales, aside from advances in the processing system between the reanalyses used, it is likely that the better performance of WFDE5 is linked to superior spatial variability of ERA5 (data


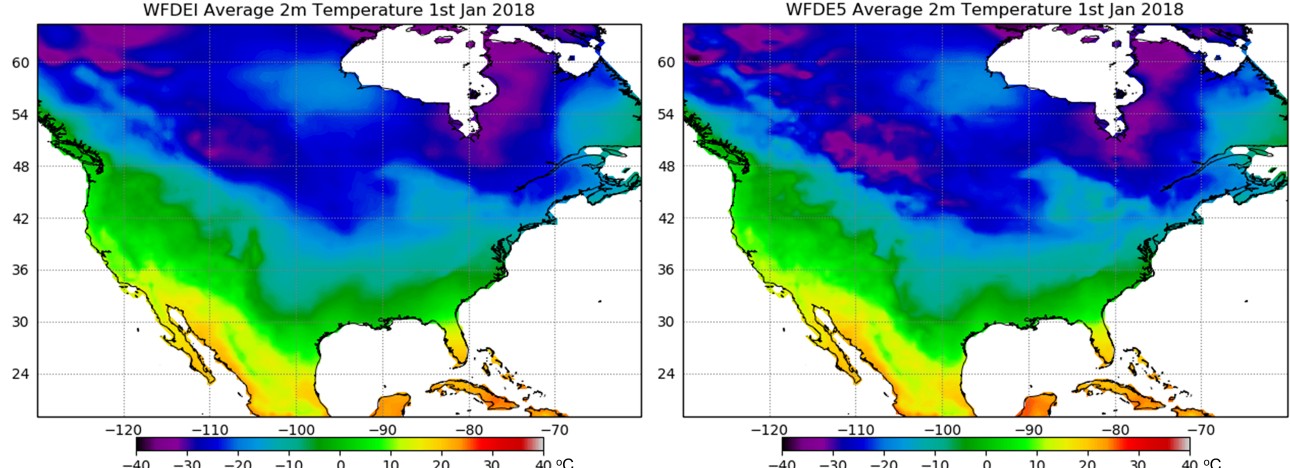

**Figure 3.** Average 2 m temperature on 1st January 2018 for north and central America.

aggregated for WFDE5) versus ERA-Interim (data interpolated for WFDEI). This can be seen in the higher-resolution features
of daily average temperature for a single day in January 2018 in north and central America in the WFDE5 data (Fig. 3).

### 4.3  Validation with a global hydrological model

Of great importance for driving impact models such as global hydrological models is the climate forcing input since the
assessment of the water balance components are highly dependent on it (Müller Schmied et al., 2016). In order to test WFDE5
in terms of suitability for use with an impact model, the global water-availability and water-use model WaterGAP (version 2.2c,
Müller Schmied et al., 2016) was used. WaterGAP calculates water storages and fluxes on global land area (except Antarctica)
on a $0.5° \times 0.5°$ resolution ($55 \times 55\,\mathrm{km}$ at the equator) and incorporates human interventions such as human water use and man-
made reservoirs. Forcing requirements are daily values for precipitation (sum of rainfall and snowfall), average temperature,
downwards shortwave radiation and downwards longwave radiation. For model specific details, the reader is referred to Müller
Schmied et al. (2016, 2014) and Döll et al. (2003). Despite the possibility of calibrating the model, WaterGAP was run with
an uncalibrated setup (model parameter $\gamma$ set to 2, whereas CFA and CFS are set to 1 globally, details can be found in Müller
Schmied et al. (2014)). This parameter choice was designated to mimic the behaviour in a typical impact model and also due
to time and technical constraints (a time series start year of 1920 or earlier is required for standard calibration). The model was
driven by ERA5, WFDE5 and WFDEI (the latter two with both the precipitation separately scaled to GPCC and CRU monthly
sums and the daily aggregation of WFDE5 (W5E5; Lange, 2019c) (see Sect. 5) was used and assessed in terms of resulting
water balance components (Table 6), for model efficiency (Fig. 4) and for river discharge seasonality for selected large river
basins (Fig. 5).

The long-term-annual water balance shows reasonably (around 10%) higher precipitation (P) for ERA5 compared to the
WFDE5 adjustments to GPCC or CRU which results in significantly higher values for actual evapotranspiration (AET) and



**Table 6.** Long-term-annual water balance components [$km^3yr^{-1}$] as simulated with uncalibrated WaterGAP 2.2c and for 1981-2010.

| No | Component | ERA5 | WFDE5-GPCC | WFDE5-CRU | WFDEI-GPCC | WFDEI-CRU |
|----|-----------|------|------------|-----------|------------|-----------|
| 1 | Precipitation | 120245 | 111529 | 110981 | 111616 | 111554 |
| 2 | Actual evapotranspiration | 76695 | 73430 | 74702 | 73540 | 74230 |
| 3 | River discharge to oceans and inland sinks | 43623 | 38135 | 36310 | 38088 | 37320 |
| 4 | Total (actual) water consumpitons (rows 5+6) | 1105 | 1183 | 1151 | 1103 | 1086 |
| 5 | Net (actual) abstraction from surface water | 1241 | 1359 | 1318 | 1246 | 1223 |
| 6 | Net abstraction from groundwater | -136 | -176 | -167 | -143 | -137 |
| 7 | Change of total water storage | -74 | -36 | -31 | -12 | 4 |
| 8 | long-term annual water balance error | 0.16 | 0.15 | 0.14 | 0.14 | 0.14 |

greater river discharge to oceans and inland sinks (Q) (Table 6). The general reduction of precipitation to observation-based datasets leads to similar values of previous estimates (e.g., Müller Schmied et al., 2014, Table 2; Müller Schmied et al., 2016) which indicate that the adjustments to precipitation rates in WFDE5 datasets are plausible. Even though WaterGAP was not

calibrated, AET and Q are well within the estimates of other models or datasets (Müller Schmied et al., 2014, Table 5) (note that compared to the NoCal variant of Müller Schmied et al. (2014) a $\gamma$ value of 2 (1 in Müller Schmied et al., 2014) was used in this case since it fits better to the original purpose of the calibration parameter). Nevertheless, the usage of the CRU and GPCC datasets to adjust ERA5 within WFDE5 (difference: 1825 $km^3yr^{-1}$) seems to have a substantial larger impact than for WFDEI (difference: 768 $km^3yr^{-1}$) which is a result of different CRU versions for adjusting WFDEI. Water consumption,

especially the net abstraction from surface water and from groundwater is around 10% higher for WFDE5 compared to WFDEI which is a result of a higher (2 $Wm^{-2}$) global average net radiation (thus larger potential evapotranspiration and consequently irrigation water demand).

The performance of the uncalibrated model runs have been assessed using the widely used Nash Sutcliffe Efficiency metric (NSE, Nash and Sutcliffe, 1970) relative to monthly time series of GRDC station observed discharge. 1216 stations have been

used out of the usual 1319 stations used for WaterGAP calibration (Müller Schmied et al., 2014) constrained by data availability for at least one year in the time span of the forcing. The optimum NSE is 1 and the value can become infinitely negative, but below 0 the simulation is not better than the average of the observations (Nash and Sutcliffe, 1970). The median performance of the model runs are similar and around the value 0 with some ranging towards optimum but also towards negative NSE values. Note that consistently around 16 to 17 % of the stations are outside of the limits of the boxplots (NSE > 1.5 * inter

quartile range) towards negative values and not displayed. Generally, the variants scaled to GPCC tend to have a slightly better performance than the values scaled to CRU. Typically, the performance increases as a result of calibrating the model (see Müller Schmied et al., 2014, Fig. 6), so the NSE values reported here should not be wrongly interpreted as the result of a poor quality of forcing data but more towards that uncalibrated impact models could reach - in principle - similar efficiencies independently of the forcing data assessed here (with slight advantages of the bias-adjusted WFDE5 data compared to direct

use of ERA5, Fig. 4).

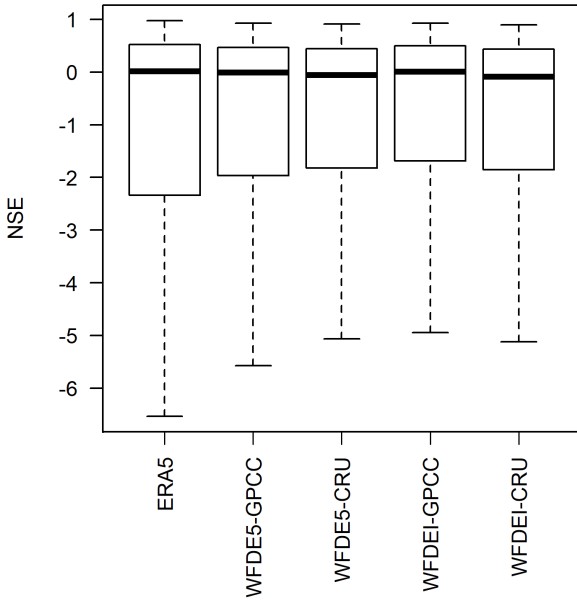

**Figure 4.** Model efficiency for the uncalibrated runs of the climate forcings in this assessment using monthly time series of 1216 GRDC stations.

In Fig. 5 discharge seasonality is shown with GRDC observations in black. The figure shows the effect of adjusting precipitation from ERA5 (red in Fig. 5). For most basins, but not for all (e.g. Mississippi), the adjustment to CRU- or GPCC-precipitation leads to a reduction of river discharge - this is substantial for some basins, e.g. Yangtze and Amazon. This does not necessarily lead to a better agreement with the observations (e.g. Amazon, Mackenzie, Lena), but for a number of basins
it does (e.g. Congo, Orange, Mekong, Danube). Interestingly, the effect of the dataset chosen to adjust precipitation (CRU v GPCC) is important for some basins (e.g. Mekong, Amazon). However, this is not relevant for other basins (e.g. Mississippi, Danube) where differences in WFDE5 and WFDEI compared to ERA5 and ERA-Interim for variables other than precipitation lead to different discharge simulations.

The validation with WaterGAP showed that using WFDE5 generally results in similar results to using WFDEI and should
be preferred to using ERA5 directly. Nevertheless, this assessment was done using uncalibrated runs, thus a proper calibration to discharge observations could highlight the full benefit of WFDE5 compared to ERA5 but this is outside of the scope of this paper.

## 5  Application in ISIMIP

WFDE5 will be employed to drive historical impact simulations and bias-adjust future climate projections in the upcoming
ISIMIP phase 3. The dataset is well suited for these purposes in particular thanks to its inter-variable consistency, which matters

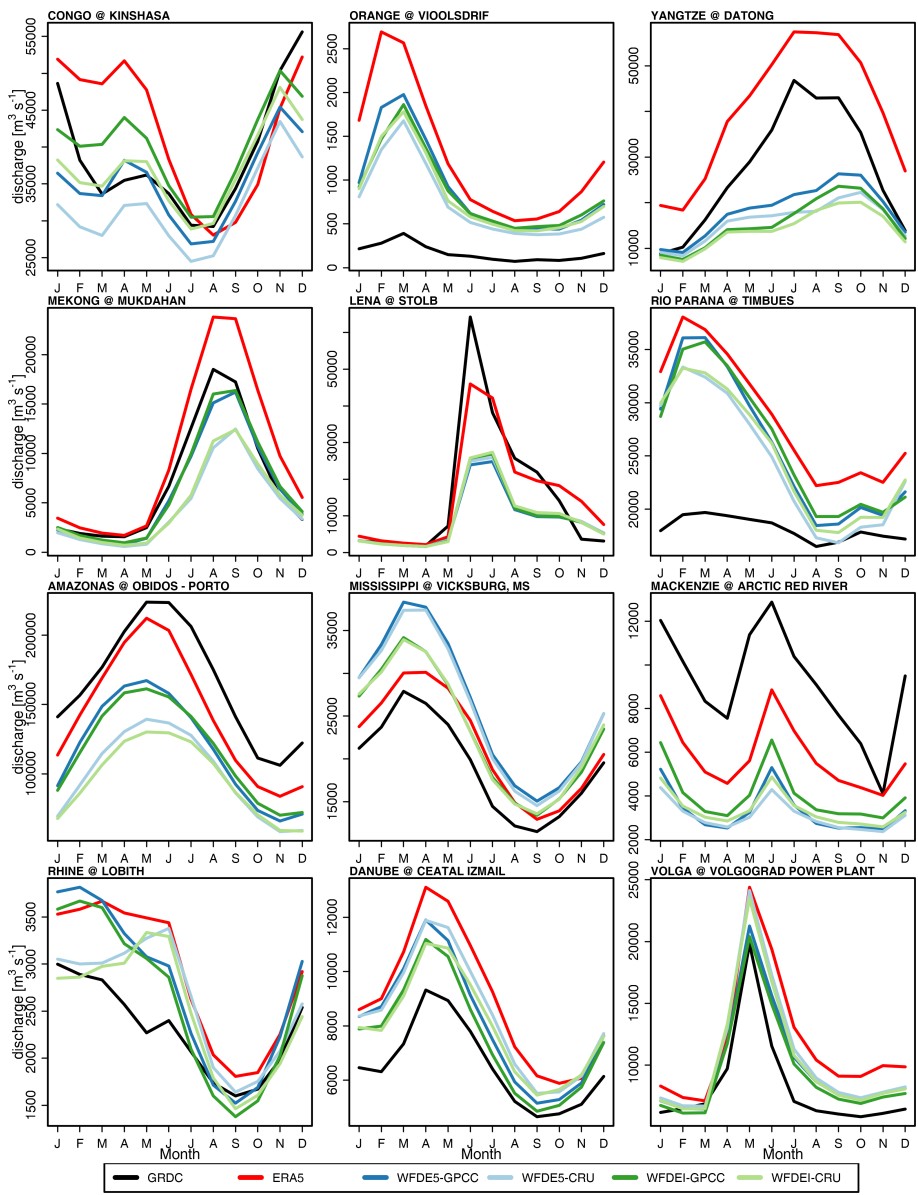

**Figure 5.** Seasonality of observed river discharge and uncalibrated WaterGAP runs for selected large river basins.

for the simulation of extreme climate impact events (Zscheischler et al., 2019). Thanks to a new bias adjustment method that is applied in ISIMIP phase 3, that is able to adjust inter-variable statistical dependencies (Lange, 2019b, 2020), the inter-variable consistency of WFDE5 will be beneficial for the bias adjustment of future climate projections as well.

Instead of using WFDE5 directly for these purposes, a derived dataset covering land and ocean with daily temporal resolution and including additional variables will be used in ISIMIP phase 3. This derived dataset is called WFDE5 over land merged





with ERA5 over the ocean (W5E5; Lange, 2019c). It covers land and ocean to facilitate impact studies everywhere and prevent mismatches between land-sea masks used by impact models and WFDE5. It has daily temporal resolution because that is sufficient to drive most impact models taking part in ISIMIP. Additional variables (2 m relative humidity, sea level pressure, total precipitation, daily maximum 2 m air temperature, daily minimum 2 m air temperature) derived from those included in
WFDE5 are included in W5E5 to meet additional impact model requirements.

W5E5 is identical to WFDE5 aggegated to daily temporal resolution where WFDE5 data are available ("over land"). Elsewhere ("over the ocean"), ERA5 data aggregated to daily temporal and 0.5° spatial resolution are used to obtain global coverage. These values over the ocean are used as is except for precipitation fluxes, which are bias-adjusted such that monthly W5E5 precipitation totals match values from version 2.3 of the Global Precipitation Climatology Project (GPCP; Adler et al., 2003).
Monthly rescaling factors used for this purpose are computed following the scale-selective rescaling procedure described by Balsamo et al. (2010).

## 6    Conclusions

The WFDE5 dataset will be useful for forcing surface models especially for near-recent hydrological and agricultural analyses. It will also be used for bias correction of the CMIP6 GCM model output in the third phase of ISIMIP. WFDE5 benefits from the
improvements of ERA5 compared to ERA-Interim. However, an advantage of WFDE5 over the direct use of surface variables from ERA5 for model forcing, are the corrections of monthly precipitation totals and adjustments to downwards shortwave to allow for cloud observations and for interannual changes in aerosol loading for some aerosol types.

WFDE5 is provided at hourly time steps versus three-hourly for WFDEI. Comparison to observations from 13 FLUXNET2015 sites distributed globally shows that, on average, WFDE5 is superior to WFDEI for all variables in terms of mean absolute er-
ror and correlation. For precipitation and wind speed WFDE5 is superior to WFDEI at all 13 sites. Although both datasets are provided at 0.5° resolution, WFDE5 has a greater spatial variability (Fig. 3) since it is obtained by aggregation of higher resolution ERA5 data rather than by interpolation of lower resolution ERA-Interim data used in WFDEI. Initial analysis using an uncalibrated hydrological model (WaterGAP) has demonstrated that the bias correction to CRU or GPCC precipitation totals results in lower discharge throughout the year bringing the global hydrological balance into better agreement with previous
studies. The corrections result in improvements towards observations relative to the use of unaltered ERA5 forcing (e.g. in the Congo, Orange and Danube basins).

Currently the WFDE5 datasets spans from the start of 1979 to the end of 2018 (end of 2016 for Rainf_WFDE5_GPCC and Snowf_WFDE5_GPCC). However, the open source Python code within the Climate Change Service Toolbox will allow users to expand the coverage back to the start of 1950 and forwards through 2019 and later for themselves. The data have been created
at 0.5° resolution to match the CRU grid, but gridded observations of precipitation totals are already available from GPCC at 0.25° and MSWEPv2 at 0.1° (Beck et al., 2019b). The future availability of gridded observations of near-surface temperature, diurnal temperature range, cloud cover, aerosol loading and numbers of wet days would allow creation of WFDE5 data at higher spatial resolution than the current dataset.





**Table A1.** Tair metrics for each FLUXNET2015 site: ERA5 v FN2015 and WFDE5 v FN2015, hourly time steps.

| Tair (°C) | No. points | Mean | MBE | | MAE | | Correlation | | Adj. Correlation | |
|---|---|---|---|---|---|---|---|---|---|---|
| | | Fluxnet 2015 | ERA5 | WFDE5 | ERA5 | WFDE5 | ERA5 | WFDE5 | ERA5 | WFDE5 |
| US-Atq | 26227 | -10.127 | 0.529 | 0.514 | 2.035 | 2.712 | 0.984 | 0.977 | 0.451 | 0.401 |
| CA-Man | 20062 | 0.741 | -1.620 | -2.138 | 1.949 | 2.452 | 0.992 | 0.988 | 0.727 | 0.712 |
| US-ARM | 23442 | 15.451 | 0.856 | 1.999 | 1.703 | 1.999 | 0.978 | 0.977 | 0.745 | 0.744 |
| BR-Sa3 | 24500 | 25.874 | 0.449 | 1.125 | 1.668 | 2.044 | 0.724 | 0.689 | 0.381 | 0.383 |
| AR-Vir | 18209 | 21.750 | -0.199 | 0.034 | 1.501 | 1.541 | 0.958 | 0.958 | 0.574 | 0.572 |
| NO-Adv | 12039 | -2.298 | -1.797 | -1.657 | 2.080 | 2.093 | 0.980 | 0.969 | 0.469 | 0.409 |
| DE-Tha | 26298 | 9.324 | -1.060 | -0.023 | 1.588 | 1.714 | 0.973 | 0.968 | 0.634 | 0.629 |
| SD-Dem | 24624 | 27.058 | 0.072 | 1.217 | 1.386 | 1.858 | 0.956 | 0.952 | 0.638 | 0.640 |
| ZA-Kru | 24825 | 21.701 | -1.424 | -0.631 | 2.194 | 2.446 | 0.887 | 0.878 | 0.425 | 0.427 |
| RU-SkP | 22123 | -2.740 | -3.891 | -4.262 | 4.276 | 4.650 | 0.984 | 0.982 | 0.331 | 0.330 |
| CN-Cng | 24117 | 6.760 | 0.321 | 0.303 | 1.320 | 1.354 | 0.994 | 0.994 | 0.526 | 0.520 |
| MY-PSO | 26295 | 25.066 | 0.753 | 1.153 | 1.155 | 1.401 | 0.884 | 0.879 | 0.593 | 0.594 |
| AU-ASM | 26239 | 22.695 | -0.269 | -0.353 | 2.082 | 2.200 | 0.950 | 0.946 | 0.502 | 0.501 |

## 7 Code and data availability

The WFDE5 dataset is distributed by the Copernicus Climate Change Service (C3S) through its Climate Data Store (CDS) and can be downloaded at https://doi.org/10.24381/cds.20d54e34 (C3S, 2020). All the CDS Toolbox scripts used to generate the WFDE5 dataset can de downloaded at the same URL. A sample of the complete dataset, which covers the whole 2016 year, is accessible without registration to the CDS at https://doi.org/10.21957/935p-cj60.

**Appendix A**





**Table A2.** PSurf metrics for each FLUXNET2015 site: ERA5 v FN2015 and WFDE5 v FN2015, hourly time steps.

| PSurf (hPa) | No. points | Mean | MBE | | MAE | | Correlation | | Adj. Correlation | |
|---|---|---|---|---|---|---|---|---|---|---|
| | | Fluxnet 2015 | ERA5 | WFDE5 | ERA5 | WFDE5 | ERA5 | WFDE5 | ERA5 | WFDE5 |
| US-Atq | 17345 | 1012.4 | -2.8 | -2.8 | 2.8 | 2.8 | 0.986 | 0.986 | 0.050 | 0.050 |
| US-ARM | 24196 | 976.3 | 0.4 | 0.5 | 0.6 | 0.6 | 0.997 | 0.996 | 0.733 | 0.656 |
| BR-Sa3 | 22274 | 986.1 | 15.0 | 15.0 | 15.0 | 15.0 | 0.880 | 0.880 | 0.051 | 0.060 |
| AR-Vir | 18048 | 999.2 | 4.1 | 4.1 | 4.1 | 4.1 | 0.992 | 0.992 | 0.883 | 0.797 |
| NO-Adv | 12038 | 1011.3 | -39.5 | -39.4 | 39.5 | 39.4 | 0.994 | 0.994 | 0.847 | 0.845 |
| DE-Tha | 26301 | 972.3 | -9.1 | -8.9 | 9.1 | 8.9 | 0.997 | 0.997 | 0.378 | 0.166 |
| SD-Dem | 6636 | 949.6 | -8.4 | -8.2 | 8.5 | 8.2 | 0.978 | 0.972 | 0.551 | 0.695 |
| RU-SkP | 22128 | 987.5 | -2.3 | -2.4 | 2.3 | 2.4 | 0.996 | 0.996 | 0.913 | 0.726 |
| CN-Cng | 24117 | 992.0 | 5.0 | 4.9 | 5.6 | 5.6 | 0.867 | 0.867 | 0.312 | 0.085 |
| AU-ASM | 26200 | 945.0 | -2.1 | -2.2 | 2.9 | 2.9 | 0.930 | 0.930 | 0.254 | 0.453 |

**Table A3.** Qair metrics for each FLUXNET2015 site: ERA5 v FN2015 and WFDE5 v FN2015, hourly time steps.

| Qair (kg kg$^{-1}$) | No. points | Mean | MBE | | MAE | | Correlation | | Adj. Correlation | |
|---|---|---|---|---|---|---|---|---|---|---|
| | | Fluxnet 2015 | ERA5 | WFDE5 | ERA5 | WFDE5 | ERA5 | WFDE5 | ERA5 | WFDE5 |
| US-Atq | 26304 | 0.00242 | -0.00010 | -0.00031 | 0.00031 | 0.00046 | 0.963 | 0.946 | 0.257 | 0.169 |
| CA-Man | 26235 | 0.00410 | -0.00049 | -0.00063 | 0.00088 | 0.00095 | 0.928 | 0.925 | 0.135 | 0.158 |
| US-ARM | 26294 | 0.00771 | 0.00026 | 0.00053 | 0.00095 | 0.00106 | 0.955 | 0.949 | 0.174 | 0.172 |
| BR-Sa3 | 26280 | 0.01537 | 0.00288 | 0.00365 | 0.00295 | 0.00370 | 0.208 | 0.201 | -0.020 | -0.033 |
| AR-Vir | 26299 | 0.01123 | 0.00056 | 0.00074 | 0.00130 | 0.00142 | 0.866 | 0.859 | 0.120 | 0.126 |
| NO-Adv | 26234 | 0.00271 | -0.00033 | -0.00035 | 0.00039 | 0.00043 | 0.979 | 0.964 | 0.228 | 0.199 |
| DE-Tha | 26304 | 0.00573 | -0.00030 | 0.00077 | 0.00049 | 0.00086 | 0.975 | 0.969 | 0.291 | 0.268 |
| SD-Dem | 26304 | 0.00800 | 0.00042 | 0.00089 | 0.00106 | 0.00134 | 0.963 | 0.959 | 0.205 | 0.192 |
| ZA-Kru | 26304 | 0.01072 | 0.00016 | 0.00079 | 0.00095 | 0.00142 | 0.922 | 0.894 | 0.169 | 0.074 |
| RU-SkP | 21505 | 0.00358 | 0.00007 | 0.00004 | 0.00054 | 0.00058 | 0.970 | 0.966 | 0.128 | 0.110 |
| CN-Cng | 26303 | 0.00486 | 0.00023 | 0.00021 | 0.00052 | 0.00051 | 0.988 | 0.987 | 0.312 | 0.301 |
| MY-PSO | 26304 | 0.01634 | 0.00157 | 0.00201 | 0.00160 | 0.00203 | 0.577 | 0.520 | 0.132 | 0.133 |
| AU-ASM | 26304 | 0.00606 | 0.00082 | 0.00083 | 0.00098 | 0.00103 | 0.962 | 0.853 | 0.218 | 0.207 |





**Table A4.** Wind metrics for each FLUXNET2015 site: ERA5 v FN2015 and WFDE5 v FN2015, hourly time steps.

| Wind (m s$^{-1}$) | No. points | Mean Fluxnet 2015 | MBE | | MAE | | Correlation | | Adj. Correlation | |
|---|---|---|---|---|---|---|---|---|---|---|
| | | | ERA5 | WFDE5 | ERA5 | WFDE5 | ERA5 | WFDE5 | ERA5 | WFDE5 |
| US-Atq | 19078 | 3.587 | 1.233 | 1.233 | 1.454 | 1.454 | 0.840 | 0.840 | 0.251 | 0.251 |
| CA-Man | 17111 | 3.403 | -0.251 | -0.251 | 0.684 | 0.684 | 0.790 | 0.790 | 0.254 | 0.254 |
| US-ARM | 21285 | 4.636 | -0.320 | -0.320 | 1.103 | 1.103 | 0.816 | 0.816 | 0.308 | 0.308 |
| BR-Sa3 | 23365 | 2.229 | -0.542 | -0.542 | 0.835 | 0.835 | 0.296 | 0.296 | 0.113 | 0.113 |
| AR-Vir | 18002 | 2.317 | 0.821 | 0.821 | 0.998 | 0.998 | 0.677 | 0.677 | 0.217 | 0.217 |
| NO-Adv | 6778 | 5.394 | -2.690 | -2.690 | 2.744 | 2.744 | 0.714 | 0.714 | 0.354 | 0.354 |
| DE-Tha | 25974 | 3.033 | -0.157 | -0.157 | 0.899 | 0.899 | 0.672 | 0.672 | 0.154 | 0.154 |
| SD-Dem | 24620 | 2.848 | 0.897 | 0.897 | 1.274 | 1.273 | 0.583 | 0.583 | 0.271 | 0.271 |
| ZA-Kru | 24825 | 3.242 | -1.160 | -1.160 | 1.309 | 1.309 | 0.645 | 0.645 | 0.241 | 0.241 |
| RU-SkP | 16790 | 2.748 | -0.042 | -0.042 | 0.616 | 0.616 | 0.786 | 0.786 | 0.182 | 0.182 |
| CN-Cng | 24096 | 3.670 | 0.080 | 0.080 | 0.904 | 0.904 | 0.821 | 0.821 | 0.305 | 0.305 |
| MY-PSO | 26061 | 1.785 | -0.303 | -0.303 | 0.769 | 0.769 | 0.317 | 0.317 | 0.019 | 0.019 |
| AU-ASM | 26213 | 2.565 | 1.289 | 1.289 | 1.430 | 1.430 | 0.675 | 0.675 | 0.322 | 0.322 |

**Table A5.** SWdown metrics for each FLUXNET2015 site: ERA5 v FN2015 and WFDE5 v FN2015, hourly time steps.

| SWdown (W m$^{-2}$) | No. points | Mean Fluxnet 2015 | MBE | | MAE | | Correlation | | Adj. Correlation | |
|---|---|---|---|---|---|---|---|---|---|---|
| | | | ERA5 | WFDE5 | ERA5 | WFDE5 | ERA5 | WFDE5 | ERA5 | WFDE5 |
| US-Atq | 25819 | 106.825 | -16.268 | -16.318 | 31.256 | 31.304 | 0.932 | 0.932 | 0.679 | 0.679 |
| CA-Man | 22867 | 132.639 | 0.883 | 0.698 | 42.677 | 42.601 | 0.916 | 0.916 | 0.612 | 0.612 |
| US-ARM | 24056 | 195.043 | 1.182 | 2.389 | 46.446 | 46.531 | 0.951 | 0.951 | 0.785 | 0.784 |
| BR-Sa3 | 24000 | 186.600 | 8.492 | 8.470 | 65.037 | 65.200 | 0.892 | 0.892 | 0.568 | 0.568 |
| AR-Vir | 15829 | 93.032 | 3.217 | 3.742 | 33.260 | 33.490 | 0.925 | 0.924 | 0.654 | 0.654 |
| NO-Adv | 12174 | 87.650 | 4.177 | 4.280 | 28.848 | 28.880 | 0.901 | 0.900 | 0.477 | 0.477 |
| DE-Tha | 26164 | 124.441 | 1.696 | 3.955 | 39.427 | 39.607 | 0.927 | 0.927 | 0.601 | 0.601 |
| SD-Dem | 24386 | 257.052 | 12.876 | 12.149 | 50.778 | 50.612 | 0.965 | 0.965 | 0.878 | 0.878 |
| ZA-Kru | 20650 | 196.204 | 2.050 | 1.279 | 53.103 | 52.882 | 0.932 | 0.932 | 0.715 | 0.715 |
| RU-SkP | 21726 | 128.763 | 1.521 | 0.161 | 45.401 | 45.304 | 0.916 | 0.916 | 0.631 | 0.631 |
| CN-Cng | 25305 | 163.944 | 9.518 | 8.084 | 41.196 | 41.016 | 0.949 | 0.949 | 0.765 | 0.765 |
| MY-PSO | 26084 | 193.032 | 0.169 | -5.169 | 62.153 | 62.170 | 0.910 | 0.909 | 0.614 | 0.613 |
| AU-ASM | 26210 | 255.772 | -3.263 | -2.342 | 84.963 | 86.203 | 0.918 | 0.917 | 0.740 | 0.740 |





**Table A6.** LWdown metrics for each FLUXNET2015 site: ERA5 v FN2015 and WFDE5 v FN2015, hourly time steps.

| LWdown (W m$^{-2}$) | No. points | Mean Fluxnet 2015 | MBE ERA5 | MBE WFDE5 | MAE ERA5 | MAE WFDE5 | Correlation ERA5 | Correlation WFDE5 | Adj. Correlation ERA5 | Adj. Correlation WFDE5 |
|---|---|---|---|---|---|---|---|---|---|---|
| US-ARM | 24106 | 335.275 | -5.157 | -1.555 | 13.372 | 13.281 | 0.964 | 0.962 | 0.342 | 0.366 |
| BR-Sa3 | 18705 | 417.810 | -3.479 | 2.234 | 10.979 | 10.591 | 0.699 | 0.729 | 0.303 | 0.336 |
| NO-Adv | 11092 | 287.082 | -26.027 | -28.481 | 30.119 | 32.124 | 0.880 | 0.868 | 0.153 | 0.159 |
| DE-Tha | 26301 | 315.195 | -7.511 | -2.397 | 15.473 | 14.744 | 0.901 | 0.898 | 0.208 | 0.216 |
| RU-SkP | 21963 | 261.041 | -17.763 | -20.020 | 22.163 | 23.999 | 0.972 | 0.970 | 0.169 | 0.182 |
| CN-Cng | 21700 | 287.760 | -11.902 | -11.579 | 14.911 | 14.822 | 0.980 | 0.980 | 0.318 | 0.325 |
| MY-PSO | 26263 | 417.346 | -2.314 | -1.719 | 11.217 | 10.728 | 0.700 | 0.722 | 0.256 | 0.277 |
| AU-ASM | 26221 | 346.663 | -5.025 | -6.203 | 10.320 | 11.483 | 0.969 | 0.964 | 0.372 | 0.377 |

**Table A7.** Precipitation (Rainf + Snowf) metrics, corrected using CRU totals, for each FLUXNET2015 site: ERA5 v FN2015 and WFDE5 v FN2015, hourly time steps.

| P. CRU (mm h$^{-1}$) | No. points | Mean Fluxnet 2015 | MBE ERA5 | MBE WFDE5 | MAE ERA5 | MAE WFDE5 | Correlation ERA5 | Correlation WFDE5 | Adj. Correlation ERA5 | Adj. Correlation WFDE5 |
|---|---|---|---|---|---|---|---|---|---|---|
| US-Atq | 26133 | 0.104 | 0.018 | 0.014 | 0.035 | 0.032 | 0.135 | 0.066 | 0.046 | 0.022 |
| CA-Man | 13634 | 0.042 | 0.047 | 0.029 | 0.097 | 0.085 | 0.232 | 0.195 | 0.079 | 0.074 |
| US-ARM | 24114 | 0.056 | 0.024 | 0.045 | 0.098 | 0.114 | 0.292 | 0.272 | 0.112 | 0.102 |
| Br-Sa3 | 26280 | 0.161 | 0.089 | 0.078 | 0.374 | 0.366 | 0.044 | 0.038 | 0.014 | 0.013 |
| NO-Adv | 6516 | 0.018 | 0.048 | 0.052 | 0.064 | 0.071 | 0.366 | 0.239 | 0.097 | 0.060 |
| DE-Tha | 26304 | 0.097 | 0.000 | -0.013 | 0.113 | 0.104 | 0.412 | 0.412 | 0.180 | 0.173 |
| SD-Dem | 24621 | 0.032 | -0.008 | 0.008 | 0.053 | 0.068 | 0.061 | 0.060 | 0.006 | 0.002 |
| ZA-Kru | 24818 | 0.044 | 0.043 | 0.033 | 0.110 | 0.101 | 0.179 | 0.174 | 0.045 | 0.038 |
| CN-Cng | 24117 | 0.037 | 0.023 | 0.017 | 0.062 | 0.059 | 0.516 | 0.455 | 0.233 | 0.194 |
| MY-PSO | 26301 | 0.220 | 0.068 | 0.029 | 0.455 | 0.420 | 0.079 | 0.075 | 0.017 | 0.014 |
| AU-ASM | 26234 | 0.032 | 0.002 | 0.001 | 0.051 | 0.055 | 0.182 | 0.159 | 0.064 | 0.072 |





**Table A8.** Precipitation (Rainf + Snowf) metrics, corrected using GPCC totals, for each FLUXNET2015 site: ERA5 v FN2015 and WFDE5 v FN2015, hourly time steps.

| P. GPCC (mm h$^{-1}$) | No. points | Mean Fluxnet 2015 | MBE ERA5 | MBE WFDE5 | MAE ERA5 | MAE WFDE5 | Correlation ERA5 | Correlation WFDE5 | Adj. Correlation ERA5 | Adj. Correlation WFDE5 |
|---|---|---|---|---|---|---|---|---|---|---|
| US-Atq | 26133 | 0.104 | 0.018 | 0.006 | 0.035 | 0.024 | 0.135 | 0.098 | 0.046 | 0.025 |
| CA-Man | 13634 | 0.042 | 0.047 | 0.042 | 0.097 | 0.094 | 0.232 | 0.220 | 0.079 | 0.079 |
| US-ARM | 24114 | 0.056 | 0.024 | 0.027 | 0.098 | 0.099 | 0.292 | 0.312 | 0.112 | 0.115 |
| BR-Sa3 | 26280 | 0.161 | 0.089 | 0.045 | 0.374 | 0.333 | 0.044 | 0.055 | 0.014 | 0.019 |
| NO-Adv | 6516 | 0.018 | 0.048 | 0.028 | 0.064 | 0.049 | 0.366 | 0.332 | 0.097 | 0.079 |
| DE-Tha | 26304 | 0.097 | 0.000 | 0.016 | 0.113 | 0.116 | 0.412 | 0.424 | 0.180 | 0.174 |
| SD-Dem | 24621 | 0.032 | -0.008 | 0.015 | 0.053 | 0.074 | 0.061 | 0.062 | 0.006 | 0.002 |
| ZA-Kru | 24818 | 0.044 | 0.043 | 0.036 | 0.110 | 0.102 | 0.179 | 0.189 | 0.045 | 0.042 |
| CN-Cng | 24117 | 0.037 | 0.023 | 0.008 | 0.062 | 0.050 | 0.516 | 0.522 | 0.233 | 0.213 |
| MY-PSO | 26301 | 0.220 | 0.068 | 0.034 | 0.455 | 0.423 | 0.079 | 0.079 | 0.017 | 0.016 |
| AU-ASM | 26234 | 0.032 | 0.002 | 0.001 | 0.051 | 0.050 | 0.182 | 0.185 | 0.064 | 0.083 |





**Table A9.** Tair metrics for each FLUXNET2015 site: WFDEI v FN2015 and WFDE5 v FN2015, 3-hourly time steps.

| Tair (°C) | No. points | Mean Fluxnet 2015 | MBE | | MAE | | Correlation | | Adj. Correlation | |
|---|---|---|---|---|---|---|---|---|---|---|
| | | | WFDEI | WFDE5 | WFDEI | WFDE5 | WFDEI | WFDE5 | WFDEI | WFDE5 |
| US-Atq | 8736 | -10.124 | 1.210 | 0.528 | 2.991 | 2.729 | 0.975 | 0.977 | 0.562 | 0.608 |
| CA-Man | 6687 | 0.741 | -2.198 | -2.086 | 2.729 | 2.453 | 0.984 | 0.987 | 0.827 | 0.830 |
| US-ARM | 7821 | 15.445 | 1.671 | 1.538 | 2.402 | 2.048 | 0.968 | 0.977 | 0.864 | 0.897 |
| BR-Sa3 | 8162 | 25.866 | 1.278 | 1.149 | 2.167 | 2.039 | 0.701 | 0.687 | 0.715 | 0.689 |
| AR-Vir | 6071 | 21.759 | 0.207 | 0.058 | 1.619 | 1.542 | 0.950 | 0.959 | 0.848 | 0.869 |
| NO-Adv | 4016 | -2.264 | -1.148 | -1.663 | 2.222 | 2.099 | 0.941 | 0.969 | 0.381 | 0.562 |
| DE-Tha | 8766 | 9.333 | -0.042 | -0.030 | 1.793 | 1.708 | 0.965 | 0.969 | 0.760 | 0.814 |
| SD-Dem | 8208 | 27.119 | 0.888 | 1.096 | 1.682 | 1.732 | 0.954 | 0.956 | 0.923 | 0.894 |
| ZA-Kru | 8274 | 21.722 | -0.710 | -0.634 | 2.266 | 2.578 | 0.909 | 0.874 | 0.832 | 0.704 |
| RU-SkP | 7373 | -2.722 | -3.572 | -4.295 | 4.192 | 4.620 | 0.982 | 0.982 | 0.705 | 0.706 |
| CN-Cng | 8035 | 6.780 | -0.027 | 0.184 | 1.977 | 1.573 | 0.986 | 0.991 | 0.778 | 0.687 |
| MY-PSO | 8761 | 25.062 | 1.372 | 1.121 | 2.055 | 1.385 | 0.711 | 0.881 | 0.662 | 0.819 |
| AU-ASM | 8743 | 22.763 | -0.468 | -0.459 | 1.534 | 1.686 | 0.971 | 0.969 | 0.905 | 0.901 |

**Table A10.** PSurf metrics for each FLUXNET2015 site: WFDEI v FN2015 and WFDE5 v FN2015, 3-hourly time steps.

| PSurf (hPa) | No. points | Mean Fluxnet 2015 | MBE | | MAE | | Correlation | | Adj. Correlation | |
|---|---|---|---|---|---|---|---|---|---|---|
| | | | WFDEI | WFDE5 | WFDEI | WFDE5 | WFDEI | WFDE5 | WFDEI | WFDE5 |
| US-Atq | 5779 | 1012.4 | -2.9 | -2.8 | 3.0 | 2.8 | 0.976 | 0.986 | 0.465 | 0.283 |
| US-ARM | 8065 | 976.3 | 0.6 | 0.5 | 1.3 | 0.6 | 0.977 | 0.996 | 0.576 | 0.929 |
| BR-Sa3 | 7451 | 986.1 | 15.3 | 15.0 | 15.3 | 15.0 | 0.576 | 0.879 | 0.421 | 0.981 |
| AR-Vir | 6022 | 999.1 | 4.0 | 4.1 | 4.0 | 4.1 | 0.966 | 0.992 | 0.265 | 0.720 |
| NO-Adv | 4015 | 1011.3 | -39.6 | -39.4 | 39.7 | 39.4 | 0.983 | 0.994 | 0.922 | 0.983 |
| DE-Tha | 8766 | 972.3 | -8.8 | -8.9 | 8.8 | 8.9 | 0.983 | 0.997 | 0.269 | 0.955 |
| SD-Dem | 2210 | 949.6 | -8.4 | -8.2 | 8.4 | 8.2 | 0.744 | 0.974 | 0.064 | 0.466 |
| RU-SkP | 7375 | 987.5 | -2.4 | -2.4 | 2.4 | 2.4 | 0.993 | 0.996 | 0.813 | 0.886 |
| CN-Cng | 8035 | 992.0 | 5.2 | 5.0 | 5.9 | 5.6 | 0.867 | 0.865 | 0.284 | 0.203 |
| AU-ASM | 8732 | 945.0 | -2.1 | -2.2 | 2.9 | 2.9 | 0.892 | 0.929 | 0.130 | 0.590 |





**Table A11.** Qair metrics for each FLUXNET2015 site: WFDEI v FN2015 and WFDE5 v FN2015, 3-hourly time steps.

| Qair (kg kg$^{-1}$) | No. points | Mean Fluxnet 2015 | MBE WFDEI | MBE WFDE5 | MAE WFDEI | MAE WFDE5 | Correlation WFDEI | Correlation WFDE5 | Adj. Correlation WFDEI | Adj. Correlation WFDE5 |
|---|---|---|---|---|---|---|---|---|---|---|
| US-Atq | 8764 | 0.00242 | -0.00004 | -0.00031 | 0.00040 | 0.00046 | 0.954 | 0.946 | 0.334 | 0.298 |
| CA-Man | 8741 | 0.00410 | -0.00077 | -0.00063 | 0.00106 | 0.00095 | 0.914 | 0.924 | 0.073 | 0.327 |
| US-ARM | 8763 | 0.00772 | -0.00038 | 0.00048 | 0.00143 | 0.00105 | 0.903 | 0.949 | 0.184 | 0.364 |
| BR-Sa3 | 8758 | 0.01538 | 0.00509 | 0.00370 | 0.00514 | 0.00375 | 0.113 | 0.215 | 0.018 | 0.070 |
| AR-Vir | 8765 | 0.01124 | -0.00007 | 0.00069 | 0.00166 | 0.00140 | 0.811 | 0.862 | 0.126 | 0.297 |
| NO-Adv | 8742 | 0.00271 | -0.00001 | -0.00035 | 0.00038 | 0.00043 | 0.946 | 0.964 | 0.186 | 0.414 |
| DE-Tha | 8767 | 0.00573 | 0.00048 | 0.00077 | 0.00077 | 0.00086 | 0.941 | 0.969 | 0.107 | 0.444 |
| SD-Dem | 8767 | 0.00800 | 0.00077 | 0.00089 | 0.00139 | 0.00134 | 0.942 | 0.959 | 0.247 | 0.387 |
| ZA-Kru | 8767 | 0.01072 | 0.00106 | 0.00077 | 0.00189 | 0.00143 | 0.852 | 0.892 | 0.078 | 0.132 |
| RU-SkP | 7186 | 0.00359 | -0.00008 | 0.00002 | 0.00061 | 0.00058 | 0.960 | 0.966 | 0.200 | 0.258 |
| CN-Cng | 8766 | 0.00486 | -0.00041 | 0.00023 | 0.00071 | 0.00053 | 0.974 | 0.987 | 0.366 | 0.497 |
| MY-PSO | 8766 | 0.01633 | 0.00216 | 0.00205 | 0.00257 | 0.00206 | 0.320 | 0.506 | 0.159 | 0.215 |
| AU-ASM | 8765 | 0.00606 | -0.00005 | 0.00084 | 0.00094 | 0.00102 | 0.921 | 0.956 | 0.090 | 0.437 |

**Table A12.** Wind metrics for each FLUXNET2015 site: WFDEI v FN2015 and WFDE5 v FN2015, 3-hourly time steps.

| Wind (m s$^{-1}$) | No. points | Mean Fluxnet 2015 | MBE WFDEI | MBE WFDE5 | MAE WFDEI | MAE WFDE5 | Correlation WFDEI | Correlation WFDE5 | Adj. Correlation WFDEI | Adj. Correlation WFDE5 |
|---|---|---|---|---|---|---|---|---|---|---|
| US-Atq | 6352 | 3.583 | 1.021 | 1.234 | 1.538 | 1.459 | 0.718 | 0.838 | 0.234 | 0.465 |
| CA-Man | 5683 | 3.404 | -1.092 | -0.251 | 1.270 | 0.687 | 0.644 | 0.792 | 0.184 | 0.504 |
| US-ARM | 7077 | 4.630 | 0.031 | -0.306 | 1.351 | 1.091 | 0.709 | 0.820 | 0.232 | 0.582 |
| BR-Sa3 | 7791 | 2.242 | -0.829 | -0.553 | 0.998 | 0.837 | 0.273 | 0.323 | 0.133 | 0.259 |
| AR-Vir | 5994 | 2.326 | 1.471 | 0.803 | 1.612 | 0.985 | 0.535 | 0.678 | 0.080 | 0.451 |
| NO-Adv | 2232 | 5.404 | -1.217 | -2.699 | 1.828 | 2.748 | 0.597 | 0.719 | 0.287 | 0.521 |
| DE-Tha | 8659 | 3.037 | 0.332 | -0.155 | 1.090 | 0.901 | 0.613 | 0.671 | 0.074 | 0.336 |
| SD-Dem | 8206 | 2.870 | 1.403 | 0.892 | 1.839 | 1.264 | 0.339 | 0.581 | 0.007 | 0.440 |
| ZA-Kru | 8274 | 3.234 | -1.580 | -1.166 | 1.676 | 1.311 | 0.446 | 0.643 | 0.190 | 0.451 |
| RU-SkP | 5599 | 2.749 | -0.732 | -0.041 | 0.946 | 0.617 | 0.642 | 0.785 | 0.176 | 0.398 |
| CN-Cng | 8025 | 3.669 | 0.415 | 0.078 | 1.175 | 0.905 | 0.736 | 0.820 | 0.321 | 0.583 |
| MY-PSO | 8682 | 1.790 | 0.368 | -0.321 | 0.951 | 0.763 | 0.129 | 0.328 | -0.054 | 0.160 |
| AU-ASM | 8736 | 2.591 | 1.935 | 1.269 | 2.110 | 1.408 | 0.439 | 0.676 | -0.080 | 0.501 |





**Table A13.** SWdown metrics for each FLUXNET2015 site: WFDEI v FN2015 and WFDE5 v FN2015, 3-hourly time steps.

| SWdown (W m$^{-2}$) | No. points | Mean Fluxnet 2015 | MBE | | MAE | | Correlation | | Adj. Correlation | |
|---|---|---|---|---|---|---|---|---|---|---|
| | | | WFDEI | WFDE5 | WFDEI | WFDE5 | WFDEI | WFDE5 | WFDEI | WFDE5 |
| US-Atq | 8560 | 107.382 | -19.644 | -16.391 | 32.138 | 29.791 | 0.936 | 0.939 | 0.893 | 0.896 |
| CA-Man | 7458 | 132.364 | 2.851 | 0.400 | 41.355 | 38.663 | 0.926 | 0.930 | 0.883 | 0.891 |
| US-ARM | 7874 | 188.303 | 5.887 | 2.023 | 49.891 | 42.040 | 0.941 | 0.958 | 0.925 | 0.940 |
| BR-Sa3 | 7691 | 188.557 | -9.459 | 8.683 | 54.996 | 54.881 | 0.916 | 0.920 | 0.902 | 0.896 |
| AR-Vir | 4676 | 104.609 | 1.621 | 4.166 | 36.737 | 34.126 | 0.925 | 0.936 | 0.874 | 0.887 |
| NO-Adv | 3920 | 86.934 | 7.180 | 4.652 | 37.221 | 25.975 | 0.862 | 0.916 | 0.710 | 0.804 |
| DE-Tha | 8659 | 124.362 | 7.860 | 3.860 | 63.740 | 34.543 | 0.857 | 0.945 | 0.759 | 0.901 |
| SD-Dem | 7961 | 256.791 | 5.962 | 12.491 | 52.392 | 47.142 | 0.964 | 0.970 | 0.955 | 0.962 |
| ZA-Kru | 6726 | 198.571 | 18.201 | 1.109 | 59.145 | 47.306 | 0.929 | 0.946 | 0.912 | 0.927 |
| RU-SkP | 6983 | 130.778 | -4.122 | -0.976 | 26.017 | 41.382 | 0.961 | 0.930 | 0.934 | 0.871 |
| CN-Cng | 8380 | 164.232 | 6.550 | 8.133 | 36.527 | 37.803 | 0.953 | 0.958 | 0.934 | 0.936 |
| MY-PSO | 8606 | 189.032 | 0.059 | -4.660 | 62.236 | 51.765 | 0.899 | 0.933 | 0.881 | 0.906 |
| AU-ASM | 8710 | 254.611 | 2.086 | -2.195 | 85.343 | 81.440 | 0.919 | 0.927 | 0.887 | 0.893 |

**Table A14.** LWdown metrics for each FLUXNET2015 site: WFDEI v FN2015 and WFDE5 v FN2015, 3-hourly time steps.

| LWdown (W m$^{-2}$) | No. points | Mean Fluxnet 2015 | MBE | | MAE | | Correlation | | Adj. Correlation | |
|---|---|---|---|---|---|---|---|---|---|---|
| | | | WFDEI | WFDE5 | WFDEI | WFDE5 | WFDEI | WFDE5 | WFDEI | WFDE5 |
| US-ARM | 8013 | 335.283 | -7.250 | -1.556 | 15.759 | 12.448 | 0.955 | 0.967 | 0.632 | 0.703 |
| BR-Sa3 | 6103 | 417.552 | 16.713 | 2.293 | 19.635 | 9.403 | 0.738 | 0.762 | 0.707 | 0.727 |
| NO-Adv | 3581 | 287.692 | -29.268 | -28.467 | 33.945 | 31.531 | 0.872 | 0.881 | 0.258 | 0.366 |
| DE-Tha | 8765 | 315.184 | -10.346 | -2.396 | 17.538 | 13.257 | 0.897 | 0.916 | 0.442 | 0.568 |
| RU-SkP | 7280 | 260.923 | -19.107 | -20.013 | 23.150 | 23.341 | 0.971 | 0.973 | 0.527 | 0.491 |
| CN-Cng | 7189 | 288.124 | -14.132 | -11.582 | 17.637 | 14.097 | 0.977 | 0.983 | 0.558 | 0.663 |
| MY-PSO | 8728 | 417.274 | -11.149 | -1.690 | 18.903 | 9.322 | 0.684 | 0.768 | 0.621 | 0.681 |
| AU-ASM | 8727 | 346.601 | -10.148 | -6.204 | 16.340 | 10.773 | 0.946 | 0.970 | 0.591 | 0.729 |



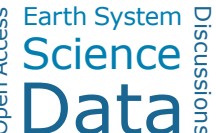

**Table A15.** Precipitation (Rainf + Snowf) metrics, corrected using CRU totals, for each FLUXNET2015 site: WFDEI v FN2015 and WFDE5 v FN2015, 3-hourly time steps.

| P. CRU (mm 3h$^{-1}$) | No. points | Mean Fluxnet 2015 | MBE WFDEI | MBE WFDE5 | MAE WFDEI | MAE WFDE5 | Correlation WFDEI | Correlation WFDE5 | Adj. Correlation WFDEI | Adj. Correlation WFDE5 |
|---|---|---|---|---|---|---|---|---|---|---|
| US-Atq | 8700 | 0.031 | 0.027 | 0.043 | 0.081 | 0.095 | 0.075 | 0.082 | 0.029 | 0.051 |
| CA-Man | 4454 | 0.122 | 0.126 | 0.092 | 0.280 | 0.236 | 0.211 | 0.282 | 0.068 | 0.209 |
| US-ARM | 8016 | 0.166 | 0.078 | 0.137 | 0.299 | 0.312 | 0.271 | 0.368 | 0.141 | 0.253 |
| Br-Sa3 | 8758 | 0.482 | 0.139 | 0.235 | 0.929 | 1.033 | 0.083 | 0.063 | 0.029 | 0.030 |
| NO-Adv | 2132 | 0.055 | 0.072 | 0.158 | 0.141 | 0.208 | 0.206 | 0.277 | 0.069 | 0.188 |
| DE-Tha | 8767 | 0.291 | 0.046 | -0.039 | 0.353 | 0.265 | 0.467 | 0.552 | 0.220 | 0.387 |
| SD-Dem | 8203 | 0.097 | 0.039 | 0.025 | 0.218 | 0.196 | 0.042 | 0.092 | 0.026 | 0.043 |
| ZA-Kru | 8266 | 0.133 | 0.119 | 0.100 | 0.317 | 0.283 | 0.223 | 0.263 | 0.105 | 0.106 |
| CN-Cng | 8036 | 0.111 | 0.032 | 0.052 | 0.168 | 0.157 | 0.483 | 0.575 | 0.307 | 0.444 |
| MY-PSO | 8764 | 0.661 | 0.103 | 0.088 | 1.239 | 1.172 | 0.078 | 0.131 | -0.012 | 0.055 |
| AU-ASM | 8739 | 0.095 | -0.001 | 0.016 | 0.149 | 0.149 | 0.182 | 0.246 | 0.052 | 0.121 |

**Table A16.** Precipitation (Rainf + Snowf) metrics, corrected using GPCC totals, for each FLUXNET2015 site: WFDEI v FN2015 and WFDE5 v FN2015, 3-hourly time steps.

| P. GPCC (mm 3h$^{-1}$) | No. points | Mean Fluxnet 2015 | MBE WFDEI | MBE WFDE5 | MAE WFDEI | MAE WFDE5 | Correlation WFDEI | Correlation WFDE5 | Adj. Correlation WFDEI | Adj. Correlation WFDE5 |
|---|---|---|---|---|---|---|---|---|---|---|
| US-Atq | 8721 | 0.031 | 0.044 | 0.018 | 0.097 | 0.071 | 0.058 | 0.127 | 0.029 | 0.070 |
| CA-Man | 4454 | 0.121 | 0.101 | 0.131 | 0.264 | 0.259 | 0.200 | 0.312 | 0.070 | 0.225 |
| US-ARM | 8016 | 0.166 | 0.122 | 0.082 | 0.337 | 0.268 | 0.255 | 0.424 | 0.138 | 0.288 |
| BR-Sa3 | 8758 | 0.482 | 0.247 | 0.135 | 1.028 | 0.936 | 0.071 | 0.088 | 0.026 | 0.048 |
| NO-Adv | 2132 | 0.055 | 0.131 | 0.085 | 0.195 | 0.142 | 0.148 | 0.385 | 0.051 | 0.268 |
| DE-Tha | 8767 | 0.291 | -0.052 | 0.047 | 0.316 | 0.295 | 0.444 | 0.569 | 0.227 | 0.395 |
| SD-Dem | 8203 | 0.097 | 0.052 | 0.046 | 0.231 | 0.214 | 0.043 | 0.095 | 0.019 | 0.042 |
| ZA-Kru | 8266 | 0.133 | 0.097 | 0.107 | 0.302 | 0.287 | 0.192 | 0.292 | 0.085 | 0.099 |
| CN-Cng | 8038 | 0.111 | 0.063 | 0.023 | 0.201 | 0.133 | 0.348 | 0.654 | 0.214 | 0.508 |
| MY-PSO | 8764 | 0.661 | 0.087 | 0.101 | 1.229 | 1.180 | 0.072 | 0.139 | -0.011 | 0.062 |
| AU-ASM | 8739 | 0.095 | 0.023 | 0.002 | 0.166 | 0.135 | 0.183 | 0.288 | 0.066 | 0.164 |





**Table A17.** Average metrics across all 13 FLUXNET2015 sites +/- 95% confidence intervals of the means for ERA5 v FN2015 and WFDE5 v FN2015 at hourly time steps (see Appendix tables A1 to A8).

| Variable | No. sites | Ave. MBE | | Ave. MAE | | Ave. Correlation | | Ave. Adj. Correlation | |
|---|---|---|---|---|---|---|---|---|---|
| | | ERA5 | WFDE5 | ERA5 | WFDE5 | ERA5 | WFDE5 | ERA5 | WFDE5 |
| Tair (°C) | 13 | -0.560 | -0.209 | 1.918 | 2.190 | 0.942 | 0.935 | 0.538 | 0.528 |
| | | ±0.820 | ±1.015 | ±0.472 | ±0.512 | ±0.045 | ±0.050 | ±0.077 | ±0.080 |
| PSurf (hPa) | 10 | -4.0 | -3.9 | 9.0 | 9.0 | 0.962 | 0.961 | 0.497 | 0.453 |
| | | ±10.2 | ±10.2 | ±8.2 | ±8.2 | ±0.036 | ±0.036 | ±0.240 | ±0.236 |
| Qair (kg kg$^{-1}$) | 13 | 0.00044 | 0.00071 | 0.00099 | 0.00163 | 0.866 | 0.853 | 0.181 | 0.160 |
| | | ±0.00055 | ±0.00068 | ±0.00042 | ±0.00053 | ±0.136 | ±0.140 | ±0.051 | ±0.049 |
| Wind (m s$^{-1}$) | 13 | -0.088 | -0.088 | 1.155 | 1.155 | 0.664 | 0.664 | 0.230 | 0.230 |
| | | ±0.646 | ±0.646 | ±0.333 | ±0.333 | ±0.107 | ±0.107 | ±0.057 | ±0.057 |
| SWdown (W m$^{-2}$) | 13 | 2.019 | 1.644 | 48.042 | 48.138 | 0.926 | 0.925 | 0.671 | 0.671 |
| | | ±7.013 | ±4.287 | ±9.415 | ±9.554 | ±0.012 | ±0.013 | ±0.064 | ±0.064 |
| LWdown (W m$^{-2}$) | 8 | -9.897 | -8.715 | 16.069 | 16.473 | 0.883 | 0.887 | 0.265 | 0.280 |
| | | ±6.900 | ±8.900 | ±5.706 | ±6.398 | ±0.099 | ±0.089 | ±0.068 | ±0.071 |
| P. CRU (mm h$^{-1}$) | 11 | 0.032 | 0.027 | 0.137 | 0.134 | 0.227 | 0.195 | 0.081 | 0.069 |
| | | ±0.048 | ±0.017 | ±0.094 | ±0.088 | ±0.103 | ±0.095 | ±0.048 | ±0.043 |
| P. GPCC (mm h$^{-1}$) | 11 | 0.032 | 0.023 | 0.137 | 0.129 | 0.227 | 0.225 | 0.081 | 0.077 |
| | | ±0.048 | ±0.010 | ±0.094 | ±0.046 | ±0.103 | ±0.105 | ±0.048 | ±0.046 |

No. sites = number of sites with measurements for each variable. Ave. = average. Adj. = Adjusted.





**Table A18.** Average metrics across all 13 FLUXNET2015 sites +/- 95% confidence intervals of the means for WFDEI v FN2015 and WFDE5 v FN2015 at 3-hourly time steps (see Appendix tables A9 to A16)

| Variable | No. sites | Ave. MBE | | Ave. MAE | | Ave. Correlation | | Ave. Adj. Correlation | |
|---|---|---|---|---|---|---|---|---|---|
| | | WFDEI | WFDE5 | WFDEI | WFDE5 | WFDEI | WFDE5 | WFDEI | WFDE5 |
| Tair (°C) | 13 | 0.310 | -0.269 | 2.279 | 2.169 | 0.923 | 0.937 | 0.762 | 0.768 |
| | | ±0.701 | ±0.982 | ±0.432 | ±0.512 | ±0.060 | ±0.051 | ±0.098 | ±0.069 |
| PSurf (hPa) | 10 | -3.9 | -3.9 | 9.2 | 9.0 | 0.896 | 0.961 | 0.421 | 0.700 |
| | | ±10.3 | ±10.2 | ±8.2 | ±8.2 | ±0.098 | ±0.037 | ±0.201 | ±0.213 |
| Qair (kg kg$^{-1}$) | 13 | 0.00060 | 0.00070 | 0.00146 | 0.00121 | 0.812 | 0.853 | 0.167 | 0.318 |
| | | ±0.00094 | ±0.00069 | ±0.00077 | ±0.00053 | ±0.164 | ±0.139 | ±0.060 | ±0.073 |
| Wind (m s$^{-1}$) | 13 | 0.117 | -0.094 | 1.414 | 1.152 | 0.525 | 0.667 | 0.137 | 0.435 |
| | | ±0.689 | ±0.646 | ±0.231 | ±0.334 | ±0.114 | ±0.103 | ±0.076 | ±0.074 |
| SWdown (W m$^{-2}$) | 13 | 1.926 | 1.637 | 48.98 | 43.604 | 0.922 | 0.939 | 0.881 | 0.901 |
| | | ±5.564 | ±4.338 | ±9.808 | ±8.530 | ±0.020 | ±0.010 | ±0.042 | ±0.023 |
| LWdown (W m$^{-2}$) | 8 | -10.586 | -8.702 | 20.363 | 15.522 | 0.880 | 0.903 | 0.542 | 0.616 |
| | | ±10.907 | ±8.905 | ±4.974 | ±6.572 | ±0.093 | ±0.077 | ±0.116 | ±0.109 |
| P. CRU (mm 3h$^{-1}$) | 11 | 0.071 | 0.082 | 0.379 | 0.373 | 0.210 | 0.266 | 0.094 | 0.172 |
| | | ±0.031 | ±0.051 | ±0.245 | ±0.247 | ±0.101 | ±0.119 | ±0.064 | ±0.095 |
| P. GPCC (mm 3h$^{-1}$) | 11 | 0.083 | 0.071 | 0.397 | 0.356 | 0.182 | 0.307 | 0.073 | 0.197 |
| | | ±0.050 | ±0.031 | ±0.249 | ±0.241 | ±0.086 | ±0.128 | ±0.063 | ±0.104 |

No. sites = number of sites with measurements for each variable. Ave. = average. Adj. = Adjusted.



*Author contributions.* MC implemented the code for the generation of the dataset, led its production and coordinated the paper's writing; GPW advised on the WATCH Forcing Data methodology and on the conversion of his WFDEI FORTRAN code into Python, he ran the validations against FLUXNET2015 site observations and checked the whole paper for consistency and English; AA contributed to the code implementation and to the production of the dataset; NB calculated the aerosol corrections; SL initiated the conversation about bias adjusting ERA5 for impact studies; HMS highlighted the need of bias adjusting ERA5 for hydrological applications at the ISIMIP workshop in Paris (June 2019) and validated the WFDE5 dataset with the global hydrological model WaterGAP; SL and HMS beta tested the WFDE5 dataset; HH provided the description of the ERA5 dataset and was involved in the discussions on the creation of the WFDE5 dataset; CB had the idea of developing WFDE5, put together the team and coordinated the different contributions. All the authors participated in the writing of the present paper, each for their own area of expertise and competence.

*Competing interests.* The authors declare that they have no conflict of interest.

*Acknowledgements.* Stefan Lange acknowledges funding from the European Union's Horizon 2020 research and innovation program under grant agreement no. 641816 (CRESCENDO). Hannes Müller Schmied is partly supported by the German Federal Ministry of Education and Research (BMBF, grant no. 01LS1711F). Marco Cucchi acknowledges partial support from the Department of Mathematics and Statistics of the University of Reading. This work used surface meteorological data collected in association with the eddy covariance data acquired and shared by the FLUXNET community, including these networks: AmeriFlux, AfriFlux, AsiaFlux, CarboAfrica, CarboEuropeIP, CarboItaly, CarboMont, ChinaFlux, Fluxnet-Canada, GreenGrass, ICOS, KoFlux, LBA, NECC, OzFlux-TERN, TCOS-Siberia, and USCCC.



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
