# Peer review of "WFDE5: bias adjusted ERA5 reanalysis data for impact studies"

_Earth System Science Data, 2020_

## Referee Comment (RC1) · Stefan Hagemann (Referee) · 26 May 2020

**Manuscript:** WFDE5: bias adjusted ERA5 reanalysis data for impact studies

**Major remarks**

The authors developed a new meteorological forcing dataset that can be used to force impact models, as reference dataset for bias correction or for climate model evaluation studies. The new WFDE5 dataset is based on bias-adjusted ERA5 reanalysis data and is a successor of the widely used WATCH forcing datasets based on ERA40 (WFD) and ERA-Interim (WFDEI). Consequently, the application potential of the WFDE5 is high and will be likely receive a similar interest by the scientific community as its two predecessors. Therefore, the dataset and the associated manuscript are well suited for a publication in ESSD.

The paper is well written and provides the necessary information about the data and includes a suitable comparison to selected Fluxnet data and to ERA5 and WFDEI data. I have only one major remark.

Currently there are only two sentences in the end of the conclusions that note the availability of 0.25° gridded precipitation datasets and the potential of utilizing the higher resolution of ERA5 instead of the present aggregation to 0.5°. This was actually my first thought about WFDE5, i.e. why it is still using 0.5° and not 0.25°? Therefore I think that the choice of losing resolution and, hence, not using 0.25° should be discussed more thoroughly with pros and cons for both resolutions. Precipitation is the most important variable and a bias adjustment with 0.25° gridded observations can already be conducted. Only using a bias adjustment of other variables with coarser resolution data (such as 0.5° CRU data) may lead to a loss of some high resolution information.

In summary, I suggest accepting the paper for publication after minor revisions are conducted.

**Minor remarks**

In the following suggestions for editorial corrections are marked in *Italic*.

Line 7
… *result* …

Line 50
ERA5 *utilizes* a vast …

Line 55
Abbreviation CMIP5 needs to be explained.

Line 132
… only *for grid*-points …

Line 181
Section 3 is largely redundant with section 7. Please remove one of these two sections.

Line 206

… of *data have* been …

Line 211
… any time *step* …

Line 277
… *performances* …

Line 307-316
It should be made clear, that W5E5 is not part of the present publication and the associated information is only provided to highlight the differences between WFDE5 and W5E5. I assume that the details of W5E5 are already published elsewhere (e.g. Lange 2019c), so the authors may even shorten this subsection.

Line 321
… shortwave *radiation* …

Line 322
Sentence is unclear and needs rewriting.

---

## Referee Comment (RC2) · Anthony Schrapffer (Referee) · 27 May 2020

**WFDE5: bias adjusted ERA5 reanalysis data for impact studies**

**Summary and General Comments**

This paper presents the WFDE5 dataset, an atmospheric forcing dataset which will further be used to drive and evaluate impact models. It is constructed by applying the WFD methodology to ERA5, the last generation of ECMWF reanalyses. WFDE5 is constructed with the same methodology as the WFDEI dataset, which has been constructed from the previous generation of ECMWF reanalyses, ERA-Interim.

WFDE5 is based on a monthly bias-correction of ERA5 by CRU TS4.03. It contains all the variables required by impact models at the conventional format (cf. ALMA conventions). In order to consider the high uncertainty of precipitation, WFDE5 is available with two different bias-adjustment of precipitation, one has been bias-adjusted by CRU and the other by GPCC. A derived daily dataset, W5E5, has also been created for upcoming ISIMIP phase 3, combining WFDE5 over the land to ERA5 over the ocean. This paper details the improvement and adjustment of the WFD method in order to fit with the ERA5 dataset.

WFDE5 benefits from the ERA5 improvements and in particular from its higher spatial and temporal resolution. Even if both WFDE5 and WFDEI have the same spatial resolution of 0.5°, the WFDE5 dataset integrates more spatial variability than WFDEI being constructed by aggregation instead of interpolation. It also integrates more temporal variability as WFDE5 is available at an hourly temporal resolution instead of 3-hourly for WFDEI The evaluation of the WFDE5 dataset is done in comparison with WFDEI and with ERA5. This is done by comparing (1) their performances over 13 FLUXNET2015 sites well distributed over the world and (2) their performance at forcing an hydrological model (WaterGAP) in order to have a first estimate of the capacity of WFDE5 to drive an hydrological model.

The manuscript is well written and well organized. This dataset is a good contribution to the land modeling community as it permits to use a bias-adjusted version of the last release of the ECMWF reanalyses, ERA5. It is promising as it will benefits from further improvement of ERA5 like, for example, from the future extensions of the period covered by ERA5. The code is publicly available so it will help the community to use the WFD method adjusted for this new version and to generate new forcings at higher resolution (when all the ground-based observation of the variables will be available at these resolutions).

I recommend the publication of this paper after some minor revisions.

**Specific comments**

**Line 127 :**
You explain how you process the grid points of CRU TS4.03 and GPCCv2018 that are not considered as land points in ERA5 and declare that "*In this way, the final WFDE5 dataset contains values only for all grid-points which are classified as land or lake by both ERA5 and CRU*".
Have you been confronted to the opposite, land points of ERA5 that are not considered as land points in CRU or GPCC ? If that was the case, how did you proceed to bias-adjust them ?

**Validation with a global hydrological model :**
The simulations with the hydrological model WaterGAP are used to assess the capacity of WFDE5 to force an hydrological model compared to ERA5 and to WFDEI. The Figure 5 shows the annual cycle of the outflow of the 12 large river basins over the period 1981-2010.
For the basins with a FLUXNET2015 stations, if we can make the hypothesis that the bias over the FLUXNET2015 stations is representative catchment area, crossing the results from the WaterGAP simulation with the previous analysis of the FLUXNET2015 stations may allow to understand which variables are responsible of the differences between CRU/GPCC and between WFDE5/WFDEI.
I think that your analysis is already quite complete but have you considered crossing the results from the WaterGAP simulations with the previous analysis of the FLUXNET2015 stations ?

**Technical corrections**

**Line 39 :**
I suggest to add the reference of ERA-40 here. The reference is present but later in the text at l. 64.

**Line 118 :**
I suppose that the "*validity date-time*" represents the start time of the time step, I suggest that you define "*validity date-time*" so the text would be clearer.

**Line 245 :**
I suggest to change: "since the assessment of the water balance components are highly dependent on it" to "since the water balance components are highly dependent on it"

**Line 255-259 :**
I think you forgot to close the parenthesis opened before "*the latter*" in this phrase :
"*The model was driven by ERA5, WFDE5 and WFDEI (the latter ...*"

**Line 267-269 :**
It is not clear that the variable you are comparing between the CRU and the GPCC version is the river discharge. I suggest to precise : "*difference of discharge : 1825 km 3 yr −1 ...*"

**Line 311 :**
Please change "*aggegated*" to "*aggregated*"

---

## Referee Comment (RC3) · Xudong Zhou (Referee) · 28 May 2020

Comments on "WFDE5: bias adjusted ERA5 reanalysis data for impact studies" by Cucchi et al.

This paper introduced the most advanced WFDE5 database for climatic usage. The performance of WFDE5 and its related WFDE5_CRU/GPCC are evaluated with site observations and spatial patterns. The results by driving a hydrological model also show improvements compared to its raw ERA5 data. In general, this paper is important for the community and should be published quickly. It is now well written and well structured. A few comments can be considered before the final acceptance.

Major comments:

[Figure]

As one of the previous referees mentioned, the ERA5 is already in 0.25o. But the authors aggregate them to 0.5 o for comparison with current 0.5o WFDEI. Some discussions should be added for this change.

Compared to ERA-I/WFDEI, ERA5/WFDE5 has superiority in small-scale weather patterns (hourly compared to 3-hourly, 0.25o/0.5 o). However, this is not shown in the results. I think, one typical event over grids or regions with storm will be helpful to show this advantage. Is it has been included in Hersbach et al. (2020, under review)?

Although the hydrological model is not the core of this paper, some of the explanations are not convincing (please see the minor comments).

Minor comments:

Line 62, 21th should be 21st

Line 88-91 & line 97-99 repeated sentences about the bias correction.

Line 97-99 how the monthly values are applied to bias correction for hourly data. What are the differences in the methods for old 3-hourly and for the hourly data here?

Figure 2. typo. a) FN205 missing '1' in FN2015 Figure 2. In table 2, rainf_CRU and rainfall_CRU+GPCC are introduced. So here Precipitation_GPCC is with CRU or not?

Table A17. Some shades in cells are useful if we compare the relative results between WFDEI and WFDE5. The cell can be with light gray if it shows a better performance. Also applicable to other Tables.

Figure 3 make a map of difference will be more straightforward for discussion. In general, the description of the spatial pattern of WFDE5 is not as solid as the discussions on point observations and later global assessment of water balance.

Line 241. No evidence shows that WFDE5 performs better without comparison to observations, even though we see the high-resolution features in WFDE5. A comparison on dense gauge observations over the US could be helpful if the author would like to do

so. (or including the topography will be helpful for the discussion on the topographic effect on temperature.) How about precipitation, its spatial characteristics could be more obvious with topographic effects.

Table 6, caption should mention that the results are for global scale.

Line 261. Not significantly higher, <5% for AET; but ∼13% for discharge.

Line 263 (previous estimates, better to use some values rather than only mentioning Table 2). Why for this comparison, in Muller Schmied et al., 2014, the STANDARD is results for WFD+WFDEI (111070), and CLIMATE scenario is WFDEI_CRU/GPCC (112969). When you do same routines to ERA5 (bias correction with CRU/GPCC), you must have this result.

Similar in Line 265 for AET and Q, some numbers are helpful for the conclusion.

Line 267-269, please specify that 1825 and 768 is for river discharge. Line 269, Differences between ERA5 and ERA-I could also lead to the difference in estimated discharge. But a more straightforward comparison is that WFDEI-CRU is 573 larger than WFDE5-CRU which can be explained by the CRU versions.

Line 270-272. Not agree. How did you explain AET in WFDE5-GPCC is less than that in WFDEI-GPCC. Abstraction goes to the river discharge rather to the AET, so I would attribute the difference to the water use schemes in hydrological model which are associated with the variabilities of forcing in ERA5 and ERA-I. If the model estimates the PET, you can list it in the Table as well.

Line 270, any reference for the net radiation difference in WFDE5 than in WFDEI (or ERA5 to ERA-I? Regarding the comparison between WFDE5 and ERA5, a global map of the differences between the precipitation and SWdown (which has been modified according to Table 2) is recommended.

Figure 5. substantial changes from ERA5 to WFDE5 in Yangtze, what has resulted in the changes?

---

## Author Comment (AC1) · 9 Jul 2020

**Replies to RC1**

*Please find below:*
- *In black, original comments by RC1*
- *In green, replies by the authors*
* * *
**Manuscript:** WFDE5: bias adjusted ERA5 reanalysis data for impact studies

The authors developed a new meteorological forcing dataset that can be used to force impact models, as reference dataset for bias correction or for climate model evaluation studies. The new WFDE5 dataset is based on bias-adjusted ERA5 reanalysis data and is a successor of the widely used WATCH forcing datasets based on ERA40 (WFD) and ERA-Interim (WFDEI). Consequently, the application potential of the WFDE5 is high and will be likely receive a similar interest by the scientific community as its two predecessors. Therefore, the dataset and the associated manuscript are well suited for a publication in ESSD.

The paper is well written and provides the necessary information about the data and includes a suitable comparison to selected Fluxnet data and to ERA5 and WFDEI data. I have only one major remark.

Currently there are only two sentences in the end of the conclusions that note the availability of 0.25° gridded precipitation datasets and the potential of utilizing the higher resolution of ERA5 instead of the present aggregation to 0.5°. This was actually my first thought about WFDE5, i.e. why it is still using 0.5° and not 0.25°? Therefore I think that the choice of losing resolution and, hence, not using 0.25° should be discussed more thoroughly with pros and cons for both resolutions. Precipitation is the most important variable and a bias adjustment with 0.25° gridded observations can already be conducted. Only using a bias adjustment of other variables with coarser resolution data (such as 0.5° CRU data) may lead to a loss of some high resolution information.

Thanks for this observation. An additional paragraph has been added to the manuscript containing a more detailed discussion on the choice of generating WFDE5 with a 0.5° x 0.5° resolution.

**2.3 Higher resolution WFDE5 data.**

The WFDE5 has been provided at 0.5° x 0.5° resolution rather than at 0.25° x 0.25° in the original ERA5 data. There are several reasons for this. The project to generate WFDE5 was designed also to deliver open source software so that users could re-generate the data at the original or, eventually, higher resolution. Three main considerations influenced the initial generation of the WFDE5 dataset:

a) The need to generate data in time for ISIMIP3 and their reporting to the AR6 of IPCC in 2020;

b) The need to convert the existing WFDEI Fortran programs into CDS Toolbox workflows and easily test the output;

c) The requirement for appropriate, and freely-available, global land gridded observations for bias correction.

The first consideration meant that any procedures adopted had to be practical and fast. The simplest way to test whether the CDS Toolbox workflows programs were working was to apply them to ERA-Interim data and check that they correctly reproduced the WFDEI data. This implied generating output at the same resolution as the WFDEI and CRU. Additionally, ISIMIP3 only required data at 0.5 x 0.5o since their models were set up at that resolution.

The WFDE5 CDS workflows will eventually allow users to generate higher resolution data on their own. At the moment, this can only be done using interpolated CRU TS4.03 and GPCCv2018 datasets, copies of which are hosted on a dedicated CDS machine and made accessible through the CDS Toolbox. Another option would be to use higher-resolution observational datasets, such as quarter-degree GPCC or MSWEP (Beck et al., 2017; 2019b) for total precipitation. This option will be viable once additional datasets can be hosted on the C3S Climate Data Store.

New reference:

Beck, H.E., Vergoploan, N., Pan, M., Levizzani, V., van Dijk, A.I.J.M., Weedon, G.P., Brocca, L., Pappenberger, F., Huffman, G.J. and Wood, E.J.: Global-scale evaluation of 22 precipitation datasets using gauge observations and hydrological modelling, Hydrology and Earth System Sciences, 21, 6201-6217, https://10.5194/hess-21-6201-2017, 2017.

In summary, I suggest accepting the paper for publication after minor revisions are conducted.

**Minor remarks**

In the following suggestions for editorial corrections are marked in *Italic*.

Line 7
... *result* …
Thanks. Done as suggested.

Line 50
ERA5 *utilizes* a vast …
Thanks. Done as suggested.

Line 55

Abbreviation CMIP5 needs to be explained.

Thanks. Done as suggested.

Line 132

... only *for grid*-points …

Thanks. Done as suggested.

Line 181

Section 3 is largely redundant with section 7. Please remove one of these two sections.

Thanks for your suggestion. Sec. 7 has been removed and merged into Sec. 3, renamed "Code and data availability".

Line 206

... of *data have* been …

Thanks. Done as suggested.

Line 211

... any time *step* …

Thanks. Done as suggested.

Line 277

... *performances* …

Thanks. Done as suggested.

Line 307-316

It should be made clear, that W5E5 is not part of the present publication and the associated information is only provided to highlight the differences between WFDE5 and W5E5. I assume that the details of W5E5 are already published elsewhere (e.g. Lange 2019c), so the authors may even shorten this subsection.

Thanks for your suggestion. Lines 311-316 have been replaced by the following sentence: "More information about the W5E5 dataset is provided by Lange et al. (2019c)."

Line 321

... shortwave *radiation* …

Thanks. Done as suggested.

Line 322

Sentence is unclear and needs rewriting.

Thanks for your suggestion. The sentence has been rephrased as follows, now connecting directly to the previous sentence: "WFDE5 benefits from the improvements of ERA5 compared to ERA-Interim as well as from the additional corrections of precipitation and shortwave radiation described above."

---

## Author Comment (AC2) · 9 Jul 2020

**Replies to RC2**

*Please find below:*
- *In black, original comments by RC2*
- *In green, replies by the authors*
* * *
**Summary and General Comments**

This paper presents the WFDE5 dataset, an atmospheric forcing dataset which will further be used to drive and evaluate impact models. It is constructed by applying the WFD methodology to ERA5, the last generation of ECMWF reanalyses. WFDE5 is constructed with the same methodology as the WFDEI dataset, which has been constructed from the previous generation of ECMWF reanalyses, ERA-Interim.

WFDE5 is based on a monthly bias-correction of ERA5 by CRU TS4.03. It contains all the variables required by impact models at the conventional format (cf. ALMA conventions). In order to consider the high uncertainty of precipitation, WFDE5 is available with two different bias-adjustment of precipitation, one has been bias-adjusted by CRU and the other by GPCC. A derived daily dataset, W5E5, has also been created for upcoming ISIMIP phase 3, combining WFDE5 over the land to ERA5 over the ocean. This paper details the improvement and adjustment of the WFD method in order to fit with the ERA5 dataset.

WFDE5 benefits from the ERA5 improvements and in particular from its higher spatial and temporal resolution. Even if both WFDE5 and WFDEI have the same spatial resolution of 0.5°, the WFDE5 dataset integrates more spatial variability than WFDEI being constructed by aggregation instead of interpolation. It also integrates more temporal variability as WFDE5 is available at an hourly temporal resolution instead of 3-hourly for WFDEI The evaluation of the WFDE5 dataset is done in comparison with WFDEI and with ERA5. This is done by comparing (1) their performances over 13 FLUXNET2015 sites well distributed over the world and (2) their performance at forcing an hydrological model (WaterGAP) in order to have a first estimate of the capacity of WFDE5 to drive an hydrological model.

The manuscript is well written and well organized. This dataset is a good contribution to the land modeling community as it permits to use a bias-adjusted version of the last release of the ECMWF reanalyses , ERA5. It is promising as it will benefits from further improvement of ERA5 like, for example, from the future extensions of the period covered by ERA5. The code is publicly available so it will help the community to use the WFD method adjusted for this new version and to generate new forcings at higher resolution (when all the ground-based observation of the variables will be available at these resolutions).

I recommend the publication of this paper after some minor revisions.

**Specific comments**

**Line 127 :**
You explain how you process the grid points of CRU TS4.03 and GPCCv2018 that are not considered as land points in ERA5 and declare that "*In this way, the final WFDE5 dataset contains values only for all grid-points which are classified as land or lake by both ERA5 and CRU*".
Have you been confronted to the opposite, land points of ERA5 that are not considered as land points in CRU or GPCC ? If that was the case, how did you proceed to bias-adjust them?
Thank you for your question. CRU and GPCC datasets have non-missing values only for grid-points considered as land. In the aforementioned case, i.e. for land points of ERA5 that are not considered as land points in CRU or GPCC, having missing values for the latter datasets would result in not being able to perform any of the described corrections. As a consequence, all grid-points which are not considered as land points in CRU and GPCC datasets are automatically set as missing values in the WFDE5 dataset.
The only exception to this regards Antarctica region, which is completely missing from CRU and GPCC datasets. As specified at the end of Section 2.2, for ERA5 land-points belonging to this region, only elevation-correction (where required) and aggregation to 0.5° x 0.5° was applied.

**Validation with a global hydrological model :**
The simulations with the hydrological model WaterGAP are used to assess the capacity of WFDE5 to force an hydrological model compared to ERA5 and to WFDEI. The Figure 5 shows the annual cycle of the outflow of the 12 large river basins over the period 1981-2010. For the basins with a FLUXNET2015 stations, if we can make the hypothesis that the bias over the FLUXNET2015 stations is representative catchment area, crossing the results from the WaterGAP simulation with the previous analysis of the FLUXNET2015 stations may allow to understand which variables are responsible of the differences between CRU/GPCC and between WFDE5/WFDEI.
I think that your analysis is already quite complete but have you considered crossing the results from the WaterGAP simulations with the previous analysis of the FLUXNET2015 stations ?
Thank you for this suggestion. Indeed, crossing the results from the meteorological and hydrological assessments would be of interest. However, we doubt this would be meaningful as the FLUXNET2015 stations represent point characteristics whereas the hydrological assessment is carried out at a much larger scale. Müller Schmied et al. (2016) assessed station measurements of radiation components with grid cell model output of WaterGAP. Even for this smaller scale gap, a scaling issue was identified (e.g. the observation stations represent in some cases a few 10 m² whereas a WaterGAP grid cell is 50x50 km at the equator). But this was solely an assessment of radiation components. For hydrological assessments, it has to be taken into consideration that hydrological models, especially those run globally, provide reasonable hydrological output for larger basins. The various uncertainties included in the model and the input data prevent a meaningful evaluation at one specific grid cell. As well as likely inconsistencies with land cover (and other physiographic input data), there are problems with: a) comparing grid cell specific

hydrological output with FLUXNET2015 (scale mismatch, no direct variable to compare with) and b) assessing the river basins where the FLUXNET2015 stations are included (FLUXNET2015 sites cannot be assumed to be representative for a whole basin). Hence, we retain the benefits of both assessments individually and do not intend to add crossing assessments. However, we do see a certain value of showing the basin outlines and FLUXNET2015 sites, hence we included the basin outlines in Fig. 1.

New caption of Fig. 1: Location of FLUXNET2015 sites used to evaluate ERA5, WFDE5 and WFDEI as well as basin outlines for the hydrological assessment.

References:
- Müller Schmied, H., Müller, R., Sanchez-Lorenzo, A., Ahrens, B. and Wild, M.: Evaluation of Radiation Components in a Global Freshwater Model with Station-Based Observations, Water, 8(10), 450, doi:10.3390/w8100450, 2016.

**Technical corrections**

**Line 39 :**
I suggest to add the reference of ERA-40 here. The reference is present but later in the text at l. 64.
Thanks. Done as suggested.

**Line 118 :**
I suppose that the "*validity date-time*" represents the start time of the time step, I suggest that you define "*validity date-time* " so the text would be clearer.
Thanks for your suggestion. Actually, the precise definition of ERA5 validity date-time depends upon the nature of each variable: for instantaneous variables, it represents the date and time at which a particular value is valid; for accumulated variables and mean rates, it represents the ending date and time of the interval over which the variable is cumulated or averaged, and hence over which each value can be considered valid.

In order to clarify this point, the paragraph starting with "They are distributed…" at line 117 and ending with "...CDS Toolbox." at line 121 has been replaced with the following:
"They are distributed at hourly resolution, and the date and time to which each value refers to is represented using the validity date/time: for instantaneous variables, it corresponds to the date and time at which each value is considered valid; for accumulated variables, it represents the ending date and time of the interval over which the variable is accumulated, and hence over which each value can be considered valid. Accumulation variables are aggregated over the hour ending at the validity date/time, and they are automatically converted to mean rates when retrieved from within the CDS Toolbox."

Furthermore Table 3 has been deleted, as it has been considered not necessary.

**Line 245 :**

I suggest to change: "since the assessment of the water balance components are highly dependent on it" to "since the water balance components are highly dependent on it"

Thanks. Done as suggested.

**Line 255-259 :**

I think you forgot to close the parenthesis opened before "*the latter*" in this phrase :

"*The model was driven by ERA5, WFDE5 and WFDEI (the latter ...*"

Thanks. Lines 255-259 have been rewritten as follows:

"The model was driven by ERA5, WFDE5 and WFDEI (the latter two with both the precipitation separately scaled to GPCC and CRU monthly sums and the daily aggregation of WFDE5 (W5E5; Lange, 2019c), see Sect. 5) and was assessed in terms of resulting water balance components (Table 6), for model efficiency (Fig. 4) and for river discharge seasonality for selected large river basins (Fig. 5)."

**Line 267-269 :**

It is not clear that the variable you are comparing between the CRU and the GPCC version is the river discharge. I suggest to precise : "*difference of discharge : 1825 km 3 yr − 1 ...*"

Thanks. Done as suggested.

**Line 311 :**

Please change "*aggegated*" to "*aggregated*"

Thanks. Done as suggested.

---

## Author Comment (AC3) · 9 Jul 2020

**Replies to RC3**

*Please find below:*
- *In black, original comments by RC3*
- *In green, replies by the authors*
* * *
**Comments on "WFDE5: bias adjusted ERA5 reanalysis data for impact studies" by Cucchi et al.**

This paper introduced the most advanced WFDE5 database for climatic usage. The performance of WFDE5 and its related WFDE5_CRU/GPCC are evaluated with site observations and spatial patterns. The results by driving a hydrological model also show improvements compared to its raw ERA5 data. In general, this paper is important for the community and should be published quickly. It is now well written and well structured. A few comments can be considered before the final acceptance.

**Major comments:**

As one of the previous referees mentioned, the ERA5 is already in 0.25o. But the authors aggregate them to 0.5 o for comparison with current 0.5o WFDEI. Some discussions should be added for this change.

Thanks for this observation. An additional paragraph has been added to the manuscript containing a more detailed discussion on the choice of generating WFDE5 with a 0.5° x 0.5° resolution.

**2.3 Higher resolution WFDE5 data.**

The WFDE5 has been provided at 0.5° x 0.5° resolution rather than at 0.25° x 0.25° in the original ERA5 data. There are several reasons for this. The project to generate WFDE5 was designed also to deliver open source software so that users could re-generate the data at the original or, eventually, higher resolution. Three main considerations influenced the initial generation of the WFDE5 dataset:

a) The need to generate data in time for ISIMIP3 and their reporting to the AR6 of IPCC in 2020;

b) The need to convert the existing WFDEI Fortran programs into CDS Toolbox workflows and easily test the output;

c) The requirement for appropriate, and freely-available, global land gridded observations for bias correction.

The first consideration meant that any procedures adopted had to be practical and fast. The simplest way to test whether the CDS Toolbox workflows programs were working was to apply them to ERA-Interim data and check that they correctly reproduced the WFDEI data. This implied generating output at the same resolution as the WFDEI and CRU. Additionally, ISIMIP3 only required data at 0.5 x 0.5o since their models were set up at that resolution.

The WFDE5 CDS workflows will eventually allow users to generate higher resolution data on their own. At the moment, this can only be done using interpolated CRU TS4.03 and GPCCv2018 datasets, copies of which are hosted on a dedicated CDS machine and made accessible through the CDS Toolbox. Another option would be to use higher-resolution observational datasets, such as quarter-degree GPCC or MSWEP (Beck et al., 2017; 2019b) for total precipitation. This option will be viable once additional datasets can be hosted on the C3S Climate Data Store.

New reference:

Beck, H.E., Vergoploan, N., Pan, M., Levizzani, V., van Dijk, A.I.J.M., Weedon, G.P., Brocca, L., Pappenberger, F., Huffman, G.J. and Wood, E.J.: Global-scale evaluation of 22 precipitation datasets using gauge observations and hydrological modelling, Hydrology and Earth System Sciences, 21, 6201-6217, https://10.5194/hess-21-6201-2017, 2017.

Compared to ERA-I/WFDEI, ERA5/WFDE5 has superiority in small-scale weather patterns (hourly compared to 3-hourly, 0.25o/0.5 o). However, this is not shown in the results. I think, one typical event over grids or regions with storm will be helpful to show this advantage. Is it has been included in Hersbach et al. (2020, under review)?

Thanks for your observation. To answer it, we added the following sentence at line 49:
'The increased level of detail of ERA5 compared to ERA-Interim has been reported in a growing number of publications. Several of these have been summarized in Hersbach et al. (2020), and the benefit of hourly resolution is illustrated for the December 1999 storm Lothar in that paper as well. Hersbach et al. (2019) shows the increased level in detail of precipitation over the North Atlantic.'

New reference:

Hersbach, H., Bell, B., Berrisford, P., Horányi, A., Sabater, J.M., Nicolas, J., Radu, R., Schepers, D., Simmons, A., Soci, C. and Dee, D., 2019. Global reanalysis: goodbye ERA-Interim, hello ERA5. ECMWF Newsl, 159, pp.17-24.

Although the hydrological model is not the core of this paper, some of the explanations are not convincing (please see the minor comments).

**Minor comments:**

Line 62, 21th should be 21st

Thanks. Done as suggested.

Line 88-91 & line 97-99 repeated sentences about the bias correction.
Thanks for this suggestion, there's a bit of overlapping between the two paragraphs indeed. Lines 86-99 will be replaced by the following:

Here we describe the WFDE5 (i.e. "WATCH Forcing Data methodology applied to ERA5 reanalysis data", C3S, 2020), a new meteorological forcing dataset for land surface and hydrological models based on the ERA5 reanalysis (Copernicus Climate Change Service, 2017). It consists of eleven variables (see Table 2) with an hourly temporal resolution on a regular longitude-latitude half-degree grid, with global spatial coverage and values defined only for land and lake points. The dataset was derived by applying the sequential elevation and monthly bias correction methods described in Weedon et al. (2010, 2011) to half-degree aggregated ERA5 reanalysis products. The monthly observational datasets used for bias correction are CRU TS4.03 from CRU (Harris et al., 2014) for 1979 to 2018 for all variables and the GPCCv2018 full data product (Schneider et al., 2018) for rainfall and snowfall rates for 1979 to 2016. In addition, as described below, the aerosol correction step for shortwave radiation has been revised with respect to WFD and WFDEI. For an outline of the methodology applied and a reference to the observation datasets used see Tables 1 and 2.

Line 97-99 how the monthly values are applied to bias correction for hourly data. What are the differences in the methods for old 3-hourly and for the hourly data here?
Thank you for your question. As mentioned at the beginning of section 2.2, the bias-correction methods used for the generation of the WFDE5 dataset are exactly the same which had been previously used for WFD and WFDEI datasets (except for Qair variable), and which are thoroughly described in Weedon et al. (2010, 2011). These methods are not impacted by the change in temporal resolution of the input datasets, so there's no significant difference to be mentioned.

References:
- Weedon, G. P., Gomes, S., Viterbo, P., Österle, H., Adam, J. C., Bellouin, N., Boucher, O., and Best, M.: The WATCH Forcing Data 1958–2001: A meteorological forcing dataset for land surface- and hydrological-models, Tech. rep., WATCH Technical Report 22, http://www.eu-watch.org/publications/technical-reports, 2010.
- Weedon, G. P., Gomes, S., Viterbo, P., Shuttleworth, W. J., Blyth, E., Österle, H., Adam, J. C., Bellouin, N., Boucher, O., and Best, M.: Creation of the WATCH Forcing Data and Its Use to Assess Global and Regional Reference Crop Evaporation over Land during the Twentieth Century, Journal of Hydrometeorology, 12, 823–848, https://doi.org/10.1175/2011JHM1369.1, 2011.

Figure 2. typo. a) FN205 missing '1' in FN2015 Figure 2. In table 2, rainf_CRU and rainfall_CRU+GPCC are introduced. So here Precipitation_GPCC is with CRU or not?
thanks for the observation. In the figure "Precipitation_GPCC" was actually meant to be "Precipitation_CRU+GPCC", so a consistent naming has been applied.

Table A17. Some shades in cells are useful if we compare the relative results between WFDEI and WFDE5. The cell can be with light gray if it shows a better performance. Also applicable to other Tables.

Thank you for this observation, but we have decided to leave the Tables as they are.

Figure 3 make a map of difference will be more straightforward for discussion. In general, the description of the spatial pattern of WFDE5 is not as solid as the discussions on point observations and later global assessment of water balance.

Thanks for your observation. Fig. 3 was actually intended to show superiority of WFDE5 versus WFDEI just in terms of spatial resolution, and as such we believe that the current figure is better suited than a map of differences. For completeness, we attach here the alternative option we considered:

[Figure]

Line 241. No evidence shows that WFDE5 performs better without comparison to observations, even though we see the high-resolution features in WFDE5. A comparison on dense gauge observations over the US could be helpful if the author would like to do so. (or including the topography will be helpful for the discussion on the topographic effect on temperature.) How about precipitation, its spatial characteristics could be more obvious with topographic effects.

Thanks for your comment. We agree that demonstrating that WFDE5 leads to better model performance requires comparison to observations. This is why we adopted a dual approach: a) assessment of the individual variables against FLUXNET2015 site observations (Section 4.2) and b) using WaterGAP to utilize all variables together and assessing the performance against observed river flows via the GRDC gauge data (Section 4.3). Extending the latter assessment to a dense gauge network across the USA is beyond the scope of this paper.

Table 6, caption should mention that the results are for global scale.

Thank you for your suggestion. In Table 6 caption, "WaterGAP 2.2c and for 1981-2010." has been replaced by "WaterGAP 2.2c for 1981-2010 and for global land area (except Antarctica and Greenland)."

Line 261. Not significantly higher, <5% for AET; but ~13% for discharge.

Thanks for your suggestion, on which we agree. In line 261 the word "significantly" has been removed.

Line 263 (previous estimates, better to use some values rather than only mentioning Table 2). Why for this comparison, in Muller Schmied et al., 2014, the STANDARD is results for WFD+WFDEI (111070), and CLIMATE scenario is WFDEI_CRU/GPCC (112969). When you do same routines to ERA5 (bias correction with CRU/GPCC), you must have this result.

Thank you for the suggestion. We intended to argue here that using ERA5 precipitation directly leads to much higher global sums compared to those global sums that are reached when monthly scaling to observation datasets (as GPCC and CRU) has been done. The references here should show only the link to those tables. But yes, indeed the numbers are not completely similar (which has to do with a different land-sea mask used in Müller Schmied et al., 2014 (plus a different GPCC version used) and also due to a different time span used within Müller Schmied et al., 2014, 2016 (1971-2000) and this study (1981-2010). Furthermore, the CLIMATE experiment in Müller Schmied et al., 2014 is using monthly climate input from CRU TS 3.2 but GPCC v6 for precipitation and a slightly different snow undercatch routine. We agree that providing all those details is not meaningful here. Hence, to not overcomplicate our message (the reduction of global mean precipitation to plausible ranges that are based on observation-based datasets), we modified this sentence to:

"The general reduction of mean global precipitation from 120000 km3/yr for ERA5 to observation-based datasets with around 111000 km3/yr for WFDE5 is consistent to previous estimates (109631 to 111050 km3/yr for the time span 1971-2000 for the snow undercatch corrected climate forcings in Table 3 and 4 of Müller Schmied et al., 2016)."

Similar in Line 265 for AET and Q, some numbers are helpful for the conclusion.

Thank you for the suggestion. We have modified the sentence in line 265 to: "...AET and Q are well within the estimates of other models or datasets (AET 62800-75981 km3/yr and Q 34400-44560 for most assessments according to Müller Schmied et al., 2014, Table 5)"
Furthermore, we have deleted the long bracket regarding the model parameter gamma (note that…) because we think that even though this is important from a model perspective, it is an unnecessary detail for the general message and does not influence the assessment.

Line 267-269, please specify that 1825 and 768 is for river discharge.

Thank you for the observation, done as suggested.

Line 269, Differences between ERA5 and ERA-I could also lead to the difference in estimated discharge. But a more straightforward comparison is that WFDEI-CRU is 573 larger than WFDE5-CRU which can be explained by the CRU versions.

Thank you for the suggestion. We agree that this part can lead to confusion and thus we have modified the sentences starting in line 267 and ending in line 269 by: "The differences in GPCC and CRU dataset versions to adjust ERA5 (ERA-Interim) precipitation for WFDE5 (WFDEI) is substantially smaller for GPCC (precipitation difference: 87 km3/yr for WFDE5-GPCC vs. WFDEI-GPCC) compared to CRU (precipitation difference: 573 km3/yr for WFDE5-CRU vs. WFDEI-CRU). Consequently, differences in simulated river discharge are higher for WFDE5-CRU vs. WFDEI-CRU (1010 km3/yr) compared to WFDE5-GPCC vs.

WFDEI-GPCC (47 km3/yr). This implies that the choice of precipitation bias adjustment target (CRU or GPCC) impacts water balance components. Water consumption ...

Line 270-272. Not agree. How did you explain AET in WFDE5-GPCC is less than that in WFDEI-GPCC. Abstraction goes to the river discharge rather to the AET, so I would attribute the difference to the water use schemes in hydrological model which are associated with the variabilities of forcing in ERA5 and ERA-I. If the model estimates the PET, you can list it in the Table as well.

Thank you for this observation. Indeed, AET is lower for WFDE5-GPCC compared to WFDEI-GPCC. Please note that AET in Table 6 contain already Total (actual) water consumptions as the water consumption can be understood as evaporated (hence "lost" water). AET differences (excluding row 4 in Table 6) are 72247 km3/yr for WFDE5-GPCC and 72437 km3/yr for WFDEI-GPCC. It is a nice idea to look at PET as well. In WaterGAP, PET is calculated by using the Priestley-Taylor algorithm with varying the alpha parameter in dependence of aridity of the grid cell (1.26 for humid grid cells and 1.74 for (semi)arid grid cells, respectively). The resulting global scale PET values are: 149880 km3/yr (ERA5), 151545 km3/yr (WFDE5-GPCC), 151428 km3/yr (WFDE5-CRU), 151104 km3/yr (WFDEI-GPCC) and 150964 km3/yr (WFDEI-CRU). Here, WFDE5-GPCC is calculated to have 441 km3/yr more PET compared to WFDEI-GPCC, which is, however translated to 190 km3/yr less AET for WFDE5-GPCC (excluding water consumption) but 80 km3/yr more actual water consumption for WFDE5-GPCC. Of course these numbers are averaging regional differences between the datasets. The inclusion of PET would add another complexity to the description which might lose the purpose of this assessment. However, we agree that it is too speculative to reduce the difference in global water consumption to the difference in global net radiation without a sufficient spatial analysis. In addition, taking into account the overall relatively small deviations between the water balance components mentioned here, we decided to delete the sentence starting in line 269.

Line 270, any reference for the net radiation difference in WFDE5 than in WFDEI (or ERA5 to ERA-I? Regarding the comparison between WFDE5 and ERA5, a global map of the differences between the precipitation and SWdown (which has been modified according to Table 2) is recommended.

Thank you for the suggestion. Based on your important comment, we re-thought our argument here. The WFDE5 only provides downward radiation fluxes. Net radiation simulated with WaterGAP depends on both, downward radiation input from the meteorological forcing and assumptions in the model about land cover-dependent emissivity and albedo. Hence, net radiation as simulated by WaterGAP cannot easily be used to evaluate WFDE5 radiation fluxes. To avoid over-complications in the assessment for the readers, we decided to delete the part with the net radiation (see our comment above). However, the suggestion to show maps with spatial differences is a very good idea which we followed. The new Fig. 1 (below) shows the differences in precipitation, whereas the Figs. A1, A2, A3 (below) show differences in shortwave downward radiation, longwave downward radiation and temperature. We believe that those figures can improve the understanding of the differences between ERA5 and WFDE5.

Figure 5. substantial changes from ERA5 to WFDE5 in Yangtze, what has resulted in the changes?

Thank you for your question. Indeed, large changes to see. For the period 1979-1988 (which is the major time series with discharge observation as plotted in the figure), basin-wide precipitation for ERA5 is 2543 km3/yr whereas for WFDE5-GPCC it is only 1779 km3/yr, which translates to discharge (AET) of 1303 (1240) for ERA5 vs. 584 (1194) for WFDE5-GPCC. Based on your earlier suggestion regarding the precipitation differences, we added Figure 1 below for differences in precipitation, plus figures for the other variables as used by WaterGAP.

[Figure]

Figure 1: Long-term (1979–2016) average precipitation of the climate forcings, displayed as absolute number for WFDE5_CRU+GPCC (a), WFDE5_CRU (b) and differences to ERA5, computed as ERA5 minus WFDE5_CRU+GPCC (c) and ERA5 minus WFDE5_CRU (d). All units in mm yr−1.

[Figure]

Figure A1: Long-term (1979–2016) average shortwave downward radiation of the climate forcings, displayed as absolute number for WFDE5 (a) and differences to ERA5, computed as ERA5 minus WFDE5 (b). All units in W m−2.

[Figure]

Figure A2: Long-term (1979–2016) average longwave downward radiation of the climate forcings, displayed as absolute number for WFDE5 (a) and differences to ERA5, computed as ERA5 minus WFDE5 (b). All units in W m−2.

[Figure]

Figure A3: Long-term (1979–2016) average temperature of the climate forcings, displayed as absolute number for WFDE5 (a) and differences to ERA5, computed as ERA5 minus WFDE5 (b). All units in °C.